# Regulatory and coding sequences of TRNP1 co-evolve with brain size and cortical folding in mammals

**Zane Kliesmete**[1†], **Lucas Esteban Wange**[1†], **Beate Vieth**[1], **Miriam Esgleas**[2,3], **Jessica Radmer**[1], **Matthias Hülsmann**[1,4,5], **Johanna Geuder**[1], **Daniel Richter**[1], **Mari Ohnuki**[1], **Magdelena Götz**[2,3,6], **Ines Hellmann**[1*‡], **Wolfgang Enard**[1*‡]

[1]Anthropology and Human Genomics, Faculty of Biology, Ludwig-Maximilians-Universität, Munich, Germany; [2]Physiological Genomics, BioMedical Center - BMC, Ludwig-Maximilians-Universität, Munich, Germany; [3]Institute for Stem Cell Research, Helmholtz Zentrum München, Germany Research Center for Environmental Health, Munich, Germany; [4]Department of Environmental Microbiology, Eawag, Dübendorf, Switzerland; [5]Department of Environmental Systems Science, ETH Zurich, Zurich, Switzerland; [6]SYNERGY, Excellence Cluster of Systems Neurology, BioMedical Center (BMC), Ludwig-Maximilians-Universität München, Munich, Germany

**\*For correspondence:**
hellmann@bio.lmu.de (IH);
enard@bio.lmu.de (WE)

[†]These authors contributed equally to this work
[‡]These authors also contributed equally to this work

**Competing interest:** The authors declare that no competing interests exist.

**Abstract** Brain size and cortical folding have increased and decreased recurrently during mammalian evolution. Identifying genetic elements whose sequence or functional properties co-evolve with these traits can provide unique information on evolutionary and developmental mechanisms. A good candidate for such a comparative approach is *TRNP1*, as it controls proliferation of neural progenitors in mice and ferrets. Here, we investigate the contribution of both regulatory and coding sequences of *TRNP1* to brain size and cortical folding in over 30 mammals. We find that the rate of TRNP1 protein evolution ($\omega$) significantly correlates with brain size, slightly less with cortical folding and much less with body size. This brain correlation is stronger than for >95% of random control proteins. This co-evolution is likely affecting TRNP1 activity, as we find that TRNP1 from species with larger brains and more cortical folding induce higher proliferation rates in neural stem cells. Furthermore, we compare the activity of putative cis-regulatory elements (CREs) of *TRNP1* in a massively parallel reporter assay and identify one CRE that likely co-evolves with cortical folding in Old World monkeys and apes. Our analyses indicate that coding and regulatory changes that increased *TRNP1* activity were positively selected either as a cause or a consequence of increases in brain size and cortical folding. They also provide an example how phylogenetic approaches can inform biological mechanisms, especially when combined with molecular phenotypes across several species.

## Editor's evaluation

This is an important paper that combines comparative analysis and experimental assays to investigate the role of protein-coding and regulatory changes at TRNP1 in mammalian brain evolution. The evidence supporting a contribution of TRNP1 is convincing, although the strength of the link between protein-coding changes and trait evolution is stronger and more readily interpretable than the data on gene regulation. The work will be of interest to researchers interested in mammalian evolution, brain evolution, and evolutionary genetics.

## Introduction

Understanding the genetic basis of complex phenotypes within and across species is central for biology. Brain phenotypes – even when as simple as size or folding – are of particular interest to many fields, because they are linked to cognitive abilities, which are of particular interest to humans (*Reader et al., 2011*; *DeCasien et al., 2022*).

Brain size and cortical folding show extensive variation across mammals, including recurrent independent increases and decreases (*Montgomery et al., 2016*; *Boddy et al., 2012*; *Lewitus et al., 2013*; *Smaers et al., 2021*). For example, most rodents have a small brain and an unfolded cortex (*Kelava et al., 2013*), while carnivores, cetaceans, and primates generally have enlarged and folded cortices, peaking in dolphin and human. Also within primates these traits vary, showing an increase on the great ape branch, but also decreases in several New World monkey species. Using comparative, that is, phylogenetic, approaches across primates and mammals, these variations have been correlated with different life history traits, such as longevity, diet, or energetic constraints (*DeCasien et al., 2017*; *DeCasien et al., 2022*; *Heldstab et al., 2022*) revealing underlying ecological factors that drive selection for larger brains.

The underlying genetic and cellular factors that are associated with these evolutionary variations in brain size and folding have not been studied across such large phylogenies. However, observational and experimental studies, especially in mice, but increasingly also in other systems like the ferret, macaques and humans, have led to major insights into the genetic and cellular mechanisms of cortical development (*Pinson and Huttner, 2021*; *Del-Valle-Anton and Borrell, 2022*; *Villalba et al., 2021*). Briefly, proliferation of neuroepithelial stem cells (NECs) that have contacts with the apical surface and basal lamina leads to the formation of the neuroepithelium during early development. NECs then become Pax6-positive apical radial glia cells (aRGCs), that continue to self-amplify before producing basal progenitors (BPs). BPs include basal radial glia cells (bRGCs) that remain Pax6 positive, loose the apical contact, and – depending on the species – can also self-amplify before eventually producing neurons. The extent of proliferation of all these neural progenitors is also influenced by their cell cycle length where a short cell cycle leads to more cycles of symmetric divisions, a delayed onset of neurogenesis, and subsequently to more neurons and a bigger cortex. Notably, proliferation of bRGCs at a particular cortical location is thought to be crucial to generate a cortical fold at this location. Hence, genes that influence the proliferation of these neural progenitors to evolutionary changes in brain size and folding.

The major focus in this respect has been on identifying and functionally characterizing genetic changes on the human or primate lineage. For example, the human-specific gene ARHGAP11B was found to induce bRGC proliferation and folding in cortices of mice, ferrets, and marmosets (*Florio et al., 2015*; *Kalebic et al., 2018*; *Heide et al., 2020*). Other examples include an amino acid substitution specific to modern humans in *TKTL1* (*Pinson et al., 2022*), human-specific NOTCH2 paralogs (*Fiddes et al., 2018*; *Suzuki et al., 2018*), the primate-specific genes TMEM14B and TBC1D3 (*Liu et al., 2017*; *Ju et al., 2016*), and an enhancer of *FZD8*, a receptor of the Wnt pathway (*Boyd et al., 2015*). While mechanistically convincing, it is unclear whether the proposed evolutionary link can be generalized as only one evolutionary lineage is investigated. Conversely, comparative approaches that correlate sequence changes with brain size changes have investigated more evolutionary lineages (*Boddy et al., 2017*; *Montgomery et al., 2016*), but these studies lack mechanistic evidence and are limited to the analysis of protein-coding regions. Here, we combine mechanistic and phylogenetic approaches to study *TRNP1*, a gene that is known to be important for cortical growth and folding by influencing aRGC and bRGC proliferation and differentiation in mice (*Stahl et al., 2013*; *Pilz et al., 2013*; *Kerimoglu et al., 2021*) and ferrets (*Martínez-Martínez et al., 2016*).

On a cellular level, expressing *Trnp1* in neural stem cells (NSCs) isolated from mouse cortices induces phase separation, accelerates mitosis, and increases proliferation (*Stahl et al., 2013*; *Esgleas et al., 2020*). Increasing *Trnp1* expression by in utero electroporation in mice and ferrets (embryonic day 13 [E13] in mice) leads to increased proliferation of aRGCs (*Stahl et al., 2013*; *Martínez-Martínez et al., 2016*). Decreasing *Trnp1* expression levels in mice or ferrets (E13) reduces aRGC proliferation, increases their differentiation into BPs, and induces cortical folding (*Stahl et al., 2013*; *Pilz et al., 2013*; *Martínez-Martínez et al., 2016*). Notably, increasing *Trnp1* expression levels by in utero electroporation at E14.5 increases bRGC proliferation (*Kerimoglu et al., 2021*) and also induces cortical folding.

Hence, *Trnp1* levels can alter proliferation and differentiation of neural progenitors and in turn alter brain size and folding in mice and ferrets. However, whether genetic changes in *TRNP1* did alter cortical size and folding during mammalian evolution is unclear. Here, we analyse the evolution of *TRNP1* regulatory and coding sequences across mammals and investigate their link to the evolution of brain size and cortical folding.

## Results

### TRNP1 amino acid substitution rates co-evolve with rates of change in brain size and cortical folding in mammals

We experimentally and computationally collected (*Camacho et al., 2009*) and aligned (*Löytynoja, 2021*) 45 mammalian TRNP1 coding sequences, including dolphin and 18 primates (99.0% completeness, *Figure 1—figure supplement 1A*). For 30 of those species, we could also compile estimates for brain size and cortical folding, as well as body mass as a potentially confounding parameter (*Figure 1A*; *Supplementary file 1c*). We quantify brain size as its weight and cortical folding as the ratio of the cortical surface over the perimeter of the brain surface, the gyrification index (GI), where a GI = 1 indicates a completely smooth brain and a GI $gt_1$ indicates higher levels of cortical folding (*Zilles et al., 1989*). This phenotypic data together with the coding sequences are the basis for our investigation in the evolutionary relation between the rate of TRNP1 protein evolution and the evolution of brain size and gyrification.

The ratio of the non-synonymous (non-neutral) and the synonymous substitution rates, $\omega$, is easily accessible and hence one of the most widespread measures of selection on protein-coding sequences, despite its limitations (*Yang, 2006*; *Nei et al., 2000*). In the absence of additional evidence, only an $\omega > 1$ can be interpreted as proof of positive selection. However, an $\omega gt_1$ requires many recurrent selective events and hence is underpowered to detect moderate amounts of positive selection. Therefore, it has become common practice to identify increases of $\omega$ on certain branches or subtrees relative to the remainder of the tree. For our question, we are analyzing the variation of $\omega$ across branches. To this end, we use the software Coevol that allows estimating the co-variance between rates of phenotypic and evolutionary sequence changes ($\omega$), while both types of information go into the optimization of branch length estimates of the underlying phylogenetic tree (*Lartillot and Poujol, 2011*). This allows to detect a correlation between the strength of selection ($\omega$) and a phenotypic trait. The question remains whether this correlation is directly caused by selection on that trait, or what we observe are indirect effects. This is not uncommon, because the strength of selection depends on the effective population size ($N_e$) of a species, which is often linked to life history traits and body size (*Ohta, 1987*; *Lynch and Walsh, 2007*). For example, species with a large body size tend to have a small $N_e$ and thus a low efficacy of selection (*Figuet et al., 2016*; *Lartillot and Poujol, 2011*). With purifying selection being the dominant force in protein sequence evolution, we would thus expect a positive correlation between $\omega$ and body size due to indirect effects of $N_e$. However, in contrast to directed selection on one trait which is targeted to specific genes, a lower efficacy in purifying selection due to $N_e$ will have an impact on all genes.

Therefore, we compiled a set of control genes in the same 30 species for which we have *TRNP1* sequences and phenotypic data. We started with all human autosomal genes that – as TRNP1 – have only one coding exon (n=1997; Human CCDS; *Pujar et al., 2018*) and a similar length (n=1088; 291–999 bp vs. 682 bp of TRNP1). For 133 (12.3%) of these we could find full-length high-quality one-to-one orthologous sequences for all 30 species (*Figure 1—figure supplement 3A*; *Supplementary file 1f*; Materials and methods). To ensure the quality of the resulting multiple sequence alignments, all of them were manually inspected. Based on the overall tree length we removed one outlier ($\sigma_{log(dS)} > 3$) leaving us with 132 control proteins that are well comparable to TRNP1 with respect to tree length, alignment quality, and $\omega$ (*Figure 1—figure supplement 3B*). Eight rather conserved genes (six with $\omega<0.04$ and two with $\omega<0.19$) did not show an acceptable parameter convergence between runs of Coevol, leaving 124 control genes well comparable to TRNP1 (*Supplementary file 1f*). If a species such as human or dolphin evolved a large, gyrified brain due to positive selection on TRNP1, we expect those lineages to show an increased rate of phenotype (brain size and GI) change and an increased $\omega$. If this pattern is consistent across the majority of branches, Coevol would infer a

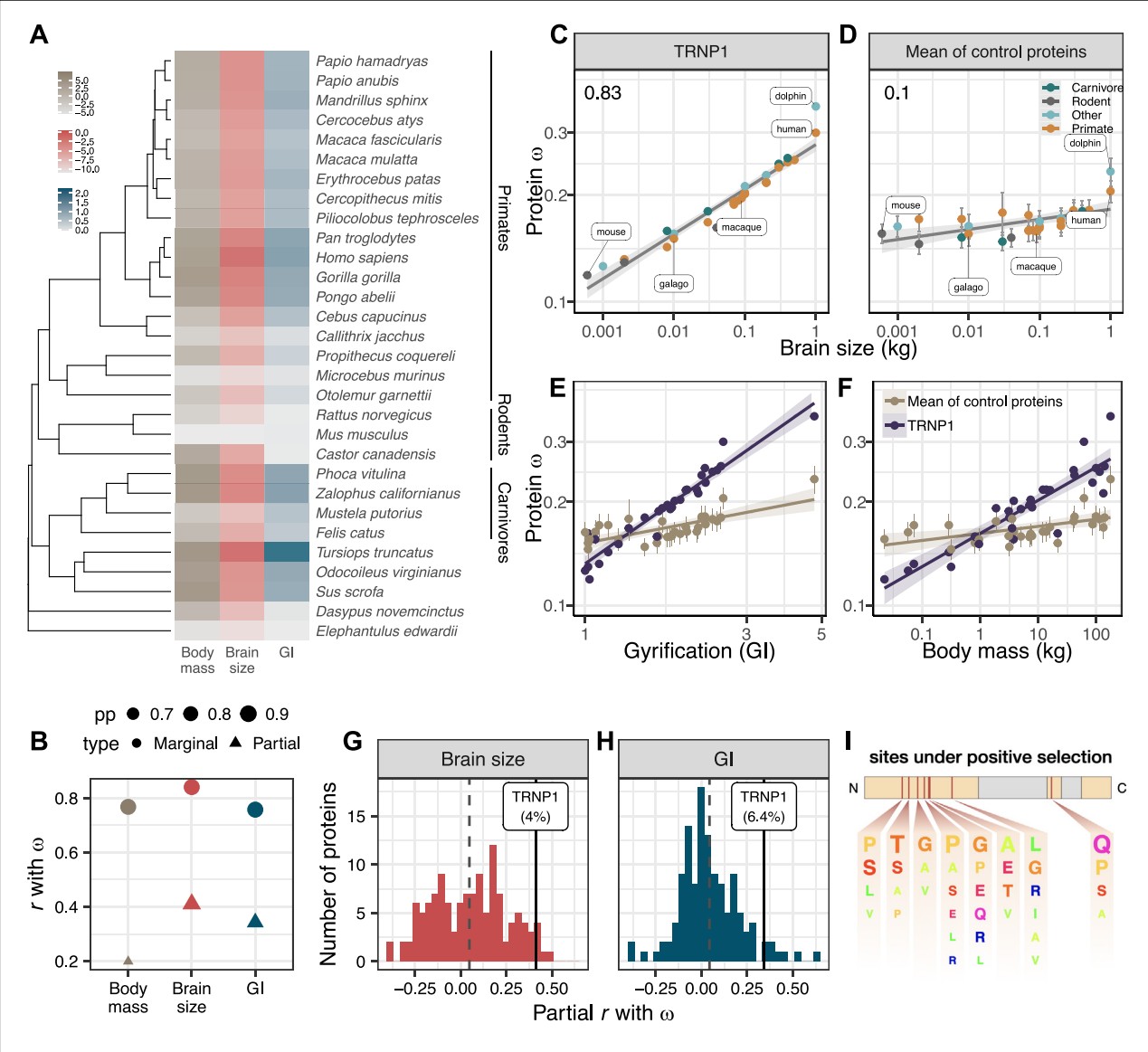

**Figure 1.** TRNP1 amino acid substitution rates co-evolve with brain size and cortical folding in mammals. (**A**) Mammalian species for which body mass, brain size, gyrification index (GI) measurements, and TRNP1 coding sequences were available (n=30)(***Figure 1—figure supplement 1***). Log2-transformed units: body mass and brain size in kg; GI is a ratio (cortical surface/perimeter of the brain surface). (**B**) Estimated marginal and partial correlation between ω of TRNP1 and the three traits using Coevol (***Lartillot and Poujol, 2011***). Size indicates posterior probability (pp). (**C**) TRNP1 protein substitution rates ($\omega$) significantly correlate with brain size ($r = 0.83$, $pp$ = 0.97).(**D**) The average correlation across 124 control proteins with brain size ($\bar{r}$=0.10). (**E**) TRNP1 ω correlation with GI compared to the average across control proteins. (**F**) TRNP1 ω correlation with body mass compared to the average across control proteins. (**C, D, E, F**) Error bars indicate standard errors. (**G**) Distribution of partial correlations between ω and brain size of the control proteins and TRNP1. (**H**) Distribution of partial correlations between ω and GI of the control proteins and TRNP1. (**I**) Scheme of the mouse TRNP1 protein (223 amino acids [AAs]) with intrinsically disordered regions (orange) and sites (red lines) subject to positive selection in mammals ($\omega > 1$, $pp > 0.95$***Figure 1—figure supplement 1***). Letter size of the depicted AAs represents the abundance of AAs at the positively selected sites.

The online version of this article includes the following figure supplement(s) for figure 1:

**Figure supplement 1.** TRNP1 protein-coding sequence analysis.

**Figure supplement 2.** Estimated marginal (**A**) and partial (**B**) correlation matrices of the combined Coevol model including the three traits and substitution rates of TRNP1.

**Figure supplement 3.** Control protein evolution rate correlation with brain size, gyrification, and body mass.

positive correlation between $\omega$ and the trait. Moreover, if this correlation is stronger than that for the average control protein, we can exclude that this is solely due to variation in the efficacy of selection.

Indeed, we find that $\omega$ of TRNP1 positively correlates with brain size (r=0.83; p=0.97), GI (r=0.75; p=0.98), and also body mass (r=0.76; p=0.97) and that these correlations are stronger than those of the average control protein (*Figure 1C–F*, *Figure 1—figure supplement 3C*), showing that the interaction between TRNP1 and the phenotypes goes beyond pure efficacy of selection effects. All three traits are highly correlated with one another. It is well known that brain and body size are not independent, and the same is true for GI and brain size (*Montgomery et al., 2016*; *Smaers et al., 2021*). To disentangle which trait is most likely to be causal for the observed correlation with $\omega$, we compare the partial correlations and find that brain size has the highest partial correlation (r=0.4), followed by GI (r=0.34), while the partial correlation with body mass (r=0.19) has a much larger drop compared to the marginals (*Figure 1B*, *Figure 1—figure supplement 3C*), making selection on brain size and/or GI the more likely causes for the variation in $\omega$. This said, TRNP1 is unlikely to be the sole evolutionary modifier of such an important and complex phenotype as brain size and gyrification. Because our control proteins represent a random selection of genes that based on sequence properties should give us comparable power to detect a link to these phenotypes, we can use the distribution of partial correlations of $\omega$ of the controls with brain size and GI to gauge the relative importance of TRNP1 for brain evolution (*Figure 1G and H*; *Supplementary file 1g*). We find that TRNP1 protein evolution is among 4.0% and 6.4% of the most correlated proteins for brain size and GI, respectively.

Having established that the rate of protein evolution of TRNP1 is linked to brain size evolution, we now want to pinpoint the relevant sites or domains in the protein to facilitate further functional studies. Using the site model of PAML (*Yang, 1997*), we find 9.8% of the codons to show signs of recurrent positive selection (i.e., $\omega > 1$, site models M8 vs. M7, $\chi^2$-value <0.001, $df$ = 2). Eight codons with a selection signature could be pinpointed with high confidence (*Supplementary file 1d*). Seven out of those eight reside within the first intrinsically disordered region (IDR) and one in the second IDR of the protein (*Figure 1I*; *Figure 1—figure supplement 1B*). The IDRs of TRNP1 are thought to mediate homotypic and heterotypic protein-protein interactions and are relevant for TRNP1-dependent phase separation, nuclear compartment size regulation, and M-phase length regulation (*Esgleas et al., 2020*). Hence, the positively selected sites indicate that these IDR-mediated TRNP1 functions were repeatedly adapted during mammalian evolution and the identified sites are candidates for further functional studies.

## TRNP1 proliferative activity co-evolves with brain size and cortical folding in mammals

Next, we investigated whether the correlation between TRNP1 protein evolution and cortical phenotypes can be linked to functional properties of TRNP1 at a cellular level. A central property of TRNP1 is to promote proliferation of aRGC (*Stahl et al., 2013*; *Esgleas et al., 2020*) and also of BPs (*Kerimoglu et al., 2021*). This proliferative activity can be assessed in an in vitro assay in which *TRNP1* is transfected into NSCs isolated from E14 mouse cortices (*Stahl et al., 2013*; *Esgleas et al., 2020*).

To compare TRNP1 orthologues in this assay, we synthesized and cloned the TRNP1 coding sequence of human, rhesus macaque, galago, mouse, and dolphin that cover the observed range of $\omega$ (*Figure 1C*). After co-transfection with green fluorescent protein (GFP), we quantified the number of proliferating (Ki67+, GFP+) over all transfected (GFP+) NSCs for each *TRNP1* orthologue in ≥7 replicates (*Figure 2A and B*). We confirmed that *TRNP1* transfection does increase proliferation compared to a GFP-only control (p-value $< 2 \times 10^{-16}$; *Figure 2—figure supplement 1A*) as shown in previous studies (*Stahl et al., 2013*; *Esgleas et al., 2020*). Remarkably, the proportion of proliferating cells was highest in cells transfected with dolphin TRNP1 followed by human, which was significantly higher than the two other primates, galago and macaque (*Figure 2C*; *Figure 2—figure supplement 1B*; *Supplementary file 2a-c*). Indeed, the proliferative activity of TRNP1 is a significant predictor for brain size (BH-adjusted p-value = 0.0018, $R^2$ = 0.89) and GI (BH-adjusted p-value = 0.016, $R^2$ = 0.69) of its species of origin (phylogenetic generalized least squares [PGLS], likelihood ratio test [LRT]; *Figure 2C*). Note that the three primates and the dolphin are phylogenetically equally distant to the mouse (*Figure 2C*) and hence a bias due to the murine assay system cannot explain the observed correlations with brain size and GI. Hence, these results further support that the TRNP1 protein co-evolves with brain size and cortical folding.

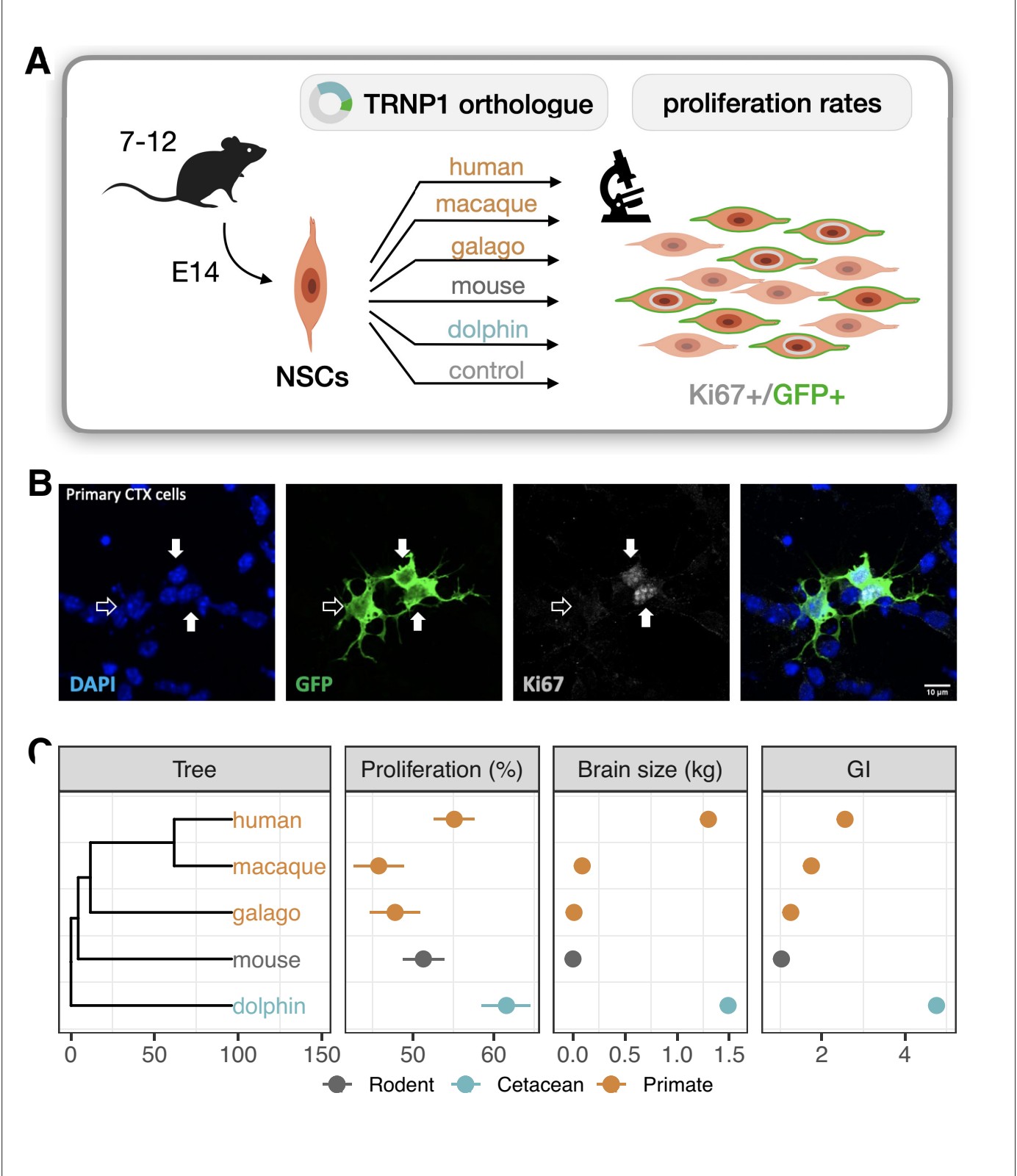

**Figure 2.** TRNP1 proliferative activity correlates with brain size and cortical folding. (**A**) Five different TRNP1 orthologues were transfected into neural stem cells (NSCs) isolated from cerebral cortices of 14-day-old mouse embryos and proliferation rates were assessed after 48 hr using Ki67 immunostaining as proliferation marker and green fluorescent protein (GFP) as transfection marker in 7–12 independent biological replicates. (**B**) Representative image of the transfected cortical NSCs immunostained for GFP and Ki67. Arrows indicate three transfected cells of which two (solid

*Figure 2 continued on next page*

*Figure 2 continued*

arrows) are Ki67-positive (*Figure 2—figure supplement 1*). (**C**) Induced proliferation in NSCs transfected with TRNP1 orthologues from five different species (*Supplementary file 2*). Proliferation rates are a significant predictor for brain size ($\chi^2$=10.04, df = 1, BH-adjusted p-value = 0.0018 = 11.75 ± 2.412, $R^2$ = 0.89) and GI ($\chi^2$=5.85, df = 1, BH-adjusted p-value = 0.016 = 16.97 ± 6.568, $R^2$ = 0.69) in the respective species (phylogenetic generalized least squares [PGLS], likelihood ratio test [LRT]). Error bars indicate standard errors. Included species: human (*Homo sapiens*), rhesus macaque (*Macaca mulatta*), northern greater galago (*Otolemur garnettii*), house mouse (*Mus musculus*), common bottlenose dolphin (*Tursiops truncatus*).

The online version of this article includes the following figure supplement(s) for figure 2:

**Figure supplement 1.** Proliferation induced by TRNP1.

## Activity of a cis-regulatory element of *TRNP1* likely co-evolves with cortical folding in catarrhines

Experimental manipulation of *Trnp1* expression levels alters proliferation and differentiation of aRGC and bRGC in mice and ferrets (*Stahl et al., 2013*; *Martínez-Martínez et al., 2016*; *Kerimoglu et al., 2021*). Therefore, we next investigated whether changes in *TRNP1* regulation may also be associated with the evolution of cortical folding and brain size by analyzing co-variation in the activity of *TRNP1* associated cis-regulatory elements (CREs), using massively parallel reporter assays (MPRAs). To this end, a library of putative regulatory sequences is cloned into a reporter vector and their activity is quantified simultaneously by the expression levels of element-specific barcodes (*Inoue and Ahituv, 2015*). To identify putative CREs of *TRNP1*, we used DNase hypersensitive sites (DHS) from human foetal brain (*Bernstein et al., 2010*) and found three upstream CREs, the promoter-including exon 1, an intron CRE, one CRE overlapping the second exon, and one downstream CRE (*Figure 3A*). We obtained the orthologous sequences of the human CREs using a reciprocal best blat (RBB) strategy across additional mammalian species either from genome databases or by sequencing, yielding a total of 351 putative CREs in a panel of 75 mammalian species (*Figure 3—figure supplement 1*).

Due to limitations in the length of oligonucleotide synthesis, we split each orthologous putative CRE into highly overlapping, 94 bp fragments. The resulting 4950 sequence tiles were synthesized together with a barcode unique for each tile. From those, we constructed a complex and unbiased lentiviral plasmid library containing at least 4251 (86%) CRE sequence tiles (*Figure 3B and C*). Next, we stably transduced this library into neural progenitor cells (NPCs) derived from two humans and one cynomolgus macaque (*Geuder et al., 2021*). We calculated the activity per CRE sequence tile as the read-normalized reporter gene expression over the read-normalized input plasmid DNA (*Figure 3A*, Materials and methods). Finally, we use the per-tile activities (*Figure 3—figure supplement 2A*) to reconstruct the activities of the putative CREs. To this end, we summed all tile sequence activities for a given CRE while correcting for the built-in sequence overlap (*Figure 3D*; Materials and methods). CRE activities correlate well within the two human NPC lines and between the human and cynomolgus macaque NPC lines, indicating that the assay is robust across replicates and species (Pearson's *r* 0.85–0.88; *Figure 3—figure supplement 2B*). The CREs covering exon 1, the intron, and the CRE downstream of *TRNP1* show the highest total activity across species while the CREs upstream of *TRNP1* show the lowest activity (*Figure 3E*).

Next, we tested whether CRE activity is associated with either brain size or GI across the 45 of the 75 mammalian species for which these phenotypes were available (*Figure 3D*). None of the CREs showed a significant association with brain size or GI (PGLS, LRT uncorrected p-value > 0.05) and only the intron CRE had a tendency to be positively associated with gyrification (PGLS, uncorrected LRT p-value=0.097, *Figure 3F*, left; *Supplementary file 3b*). Our power to detect such associations might be considerably lower than for coding sequences also because regulatory elements have a high turn-over rate (*Danko et al., 2018*; *Berthelot et al., 2018*; *Huber et al., 2020*). Hence, we expect that some orthologous DNA sequences that are CREs in one species do not function as CREs in others and can even be lost. The latter effect might explain why the sequences orthologous to human CREs are shorter in non-primate species more distantly related to humans (*Figure 3—figure supplement 1*). So phylogenetic comparisons of regulatory elements might be more powerful when restricted to species closely related to the species from which the CRE annotation is derived (humans in our case). Indeed, when we restrict our analysis to the catarrhine clade that encompasses Old World monkeys, great apes, and humans, the association between intron CRE activity and GI becomes considerably stronger (PGLS, uncorrected LRT p-value=0.003, Bonferroni-corrected for seven regions

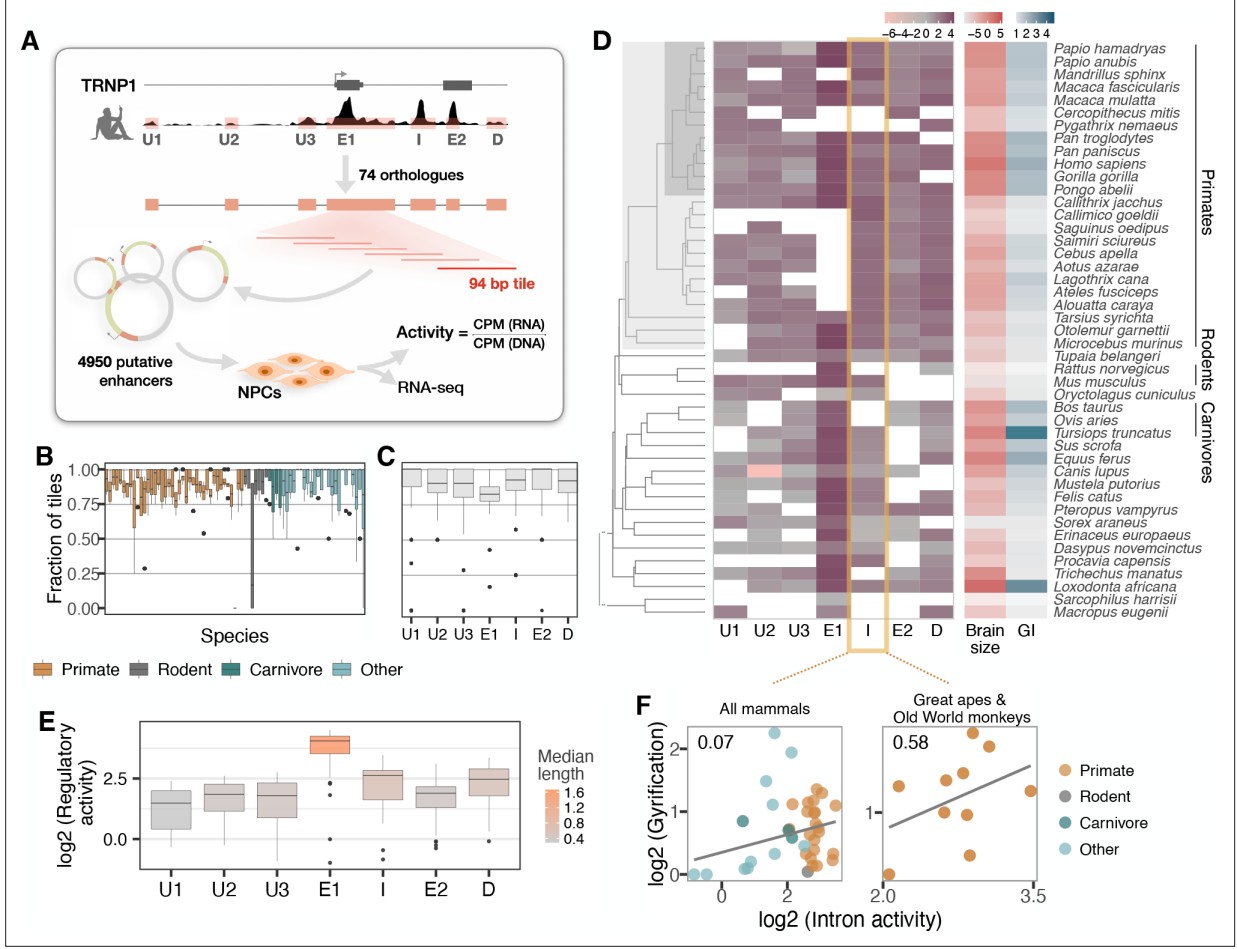

**Figure 3.** Activity of a cis-regulatory element (CRE) of *TRNP1* correlates with cortical folding in catarrhines. (**A**) Experimental setup of the massively parallel reporter assay (MPRA). Regulatory activity of seven putative TRNP1 CREs from 75 species were assayed in neural progenitor cells (NPCs) derived from human and cynomolgus macaque induced pluripotent stem cells. (*Figure 3—figure supplement 1*). (**B**) Fraction of the detected CRE tiles in the plasmid library per species across regions. The detection rates are unbiased and uniformly distributed across species and clades with only one extreme outlier *Dipodomys ordii*. (**C**) Fraction of the detected CRE tiles in the plasmid library per region across species. (**D**) Log-transformed total regulatory activity per CRE in human NPCs across species with available brain size and gyrification index (GI) measurements (n=45). (**E**) Total activity per CRE across species. Exon 1 (E1), intron (I), and the downstream (D) regions are more active and longer than other regions. (**B, C, E**) Each box represents the median and first and third quartiles with the whiskers indicating the furthest value no further than 1.5 * IQR from the box. Individual points indicate outliers. *Figure 3—figure supplement 2* (**F**) Regulatory activity of the intron CRE is weakly associated with gyrification across mammals (phylogenetic generalized least squares [PGLS], likelihood ratio test [LRT] p-value=0.097, $R^2$=0.07, n=37) and strongest across great apes and Old World monkeys, that is, catarrhines (PGLS, LRT p-value=0.003, $R^2$=0.58, n=10).

The online version of this article includes the following figure supplement(s) for figure 3:

**Figure supplement 1.** Length of the covered cis-regulatory element (CRE) sequences in the massively parallel reporter assay (MPRA) library across the tree.

**Figure supplement 2.** Analysis of massively parallel reporter assay (MPRA) data.

p-value=0.02, *Figure 3F*, right; *Supplementary file 3*). To validate that our model results are rather specific, we generated a null distribution for the observed correlation across catarrhines, permuting the activities of all other CREs of this study. In agreement with our model results, we find 8/1000 (0.8%) of the random CRE combinations to have such a significant association of p ≤ 0.003. Moreover, the intron CRE activity-GI association was consistently detected across all three cell lines including the cynomolgus macaque NPCs (*Supplementary file 3*). Furthermore, Reilly et al. compared enhancer activity by histone modifications in the developing cortex of humans, rhesus macaques, and mice and found a gain in activity on the human lineage in a region overlapping the intron CRE (*Reilly et al., 2015*). Thus, while the statistical evidence from our MPRA data alone is limited, we consider the

GI association in catarrhines together with the additional evidence from *Reilly et al., 2015*, strong enough to warrant a more detailed analysis of the intron CRE.

## Transcription factors with binding site enrichment on intron CREs regulate cell proliferation and are candidates to explain the observed activity across catarrhines

Reasoning that differences in CRE activities will likely be mediated by differences in their interactions with transcription factors (TF), we analysed the sequence evolution of putative TF binding sites (*Figure 4A*). First, we performed RNA-seq on the same samples that were used for the MPRA. Notably, also *TRNP1* was expressed (*Figure 4B*), supporting the relevance of our cellular system. Moreover, *TRNP1* expression was significantly higher in human NPC lines than that of cynomolgus macaque's (BH-adjusted p-value <0.05, *Figure 4—figure supplement 1A–C*), consistent with higher intron CRE activity. Among the 392 expressed TFs with known binding motifs, we identified 22 with an excess of binding sites (*Frith et al., 2003*) within the catarrhine intron CRE sequences (*Figure 4B and D*). In agreement with TRNP1 itself being involved in the regulation of cell proliferation (*Volpe et al., 2006*; *Stahl et al., 2013*; *Esgleas et al., 2020*), these 22 TFs are enriched in biological processes regulating cell proliferation, neuron apoptotic process, and hormone levels (Gene Ontology, Fisher's exact p-value <0.05, background: 392 expressed TFs; *Figure 4C*; *Supplementary file 3*).

To further prioritize these 22 TFs, we used the motif binding scores in the 10 catarrhine intron CREs to predict the observed intron CRE activity in the MPRA and to predict the GI of the respective species. We found three TFs (CTCF, ZBTB26, SOX8) to be the best candidates to explain the variation in the intron CRE activity and one TF (CTCF) to co-vary with GI (PGLS, uncorrected LRT p-value <0.05, *Figure 4D–F*). While the statistical support for this association is not strong, which is expected given that we were screening 22 candidate TFs in only 10 species, CTCF ChIP-seq data from the relevant cell types suggests that this particular CTCF binding site is indeed bound by CTCF in human NPCs (ChiP-seq, *Encode Project Consortium, 2012*, *Figure 4—figure supplement 2*). Moreover, HiC data show a topologically associated domain (TAD) boundary just upstream of *TRNP1* in the germinal zone of the developing human brain (postconception week 8, *Won et al., 2016*). Hence, variations in the binding strength of CTCF across species might likely have consequences for the stability of the TAD boundary and *TRNP1* expression, affecting the associated phenotypes given its crucial role for brain development (*Stahl et al., 2013*).

In summary, we find a suggestive correlation between the activity of the intron CRE and gyrification in catarrhines, indicating that also regulatory changes of *TRNP1* might have contributed to the evolution of gyrification.

## Discussion

Previous studies in mice and ferrets have elucidated mechanisms how Trnp1 is necessary for proliferation and differentiation of neural progenitors and how it could contribute to the evolution of brain size and cortical folding. We applied phylogenetic methods to explore associations between sequence and trait evolution and found that the rate of protein evolution and the proliferative activity of TRNP1 positively correlate with brain size and gyrification in mammals. Moreover, we find tentative evidence that the activity of a regulatory element in the intron of *TRNP1* might be associated with gyrification in catarrhines. At the sequence level, such a correlation could also be caused by confounding factors that affect the efficacy of natural selection such as the effective population size (*Ohta, 1987*; *Lynch and Walsh, 2007*). However, body size – a reasonable proxy for effective population size (*Figuet et al., 2016*; *Lartillot and Poujol, 2011*) – correlates much less with TRNP1 protein evolution than brain size or gyrification. Even more convincingly, the correlation of TRNP1 with brain size and gyrification is much stronger than the average correlation of these traits with the evolution of other proteins, that would have had to experience the same population size changes. Furthermore, it is unclear how an increased proliferative activity of TRNP1 or an increased CRE activity could be caused by a reduced efficacy of selection or other confounding factors. Together with the known role of TRNP1 in brain development, we think that the observed correlations are best interpreted as co-evolution of TRNP1 activity with brain size and gyrification, that is, that more active TRNP1 alleles were selected because they were advantageous to increase brain size and/or gyrification.

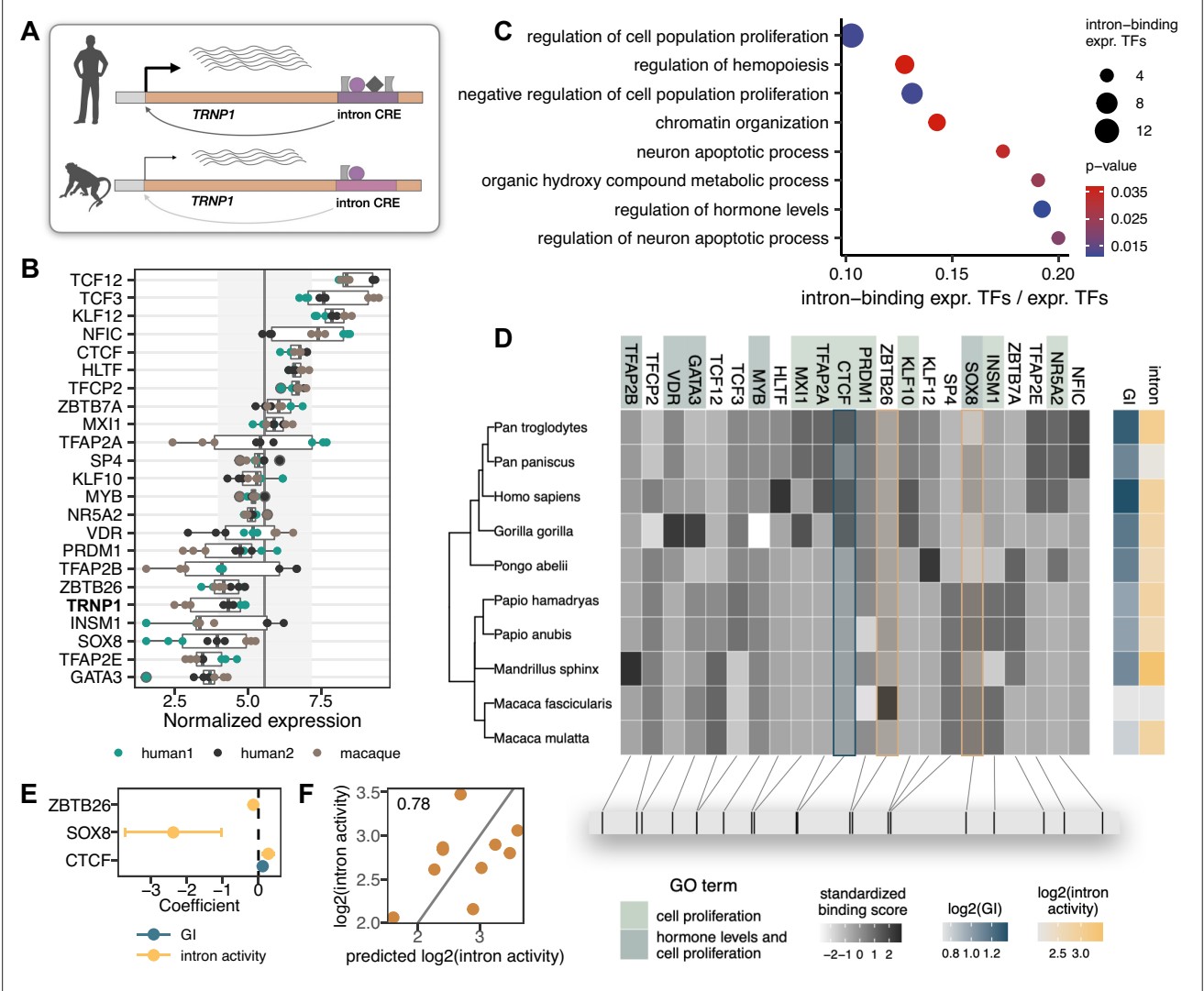

**Figure 4.** Transcription factors (TFs) with binding site enrichment on intron cis-regulatory elements (CREs) regulate cell proliferation and are candidates to explain the observed activity across catarrhines. (**A**) Orthologous intron CRE sequences show different regulatory activities under the same cellular conditions, suggesting variation in cis regulation across species. (**B**) Variance-stabilized expression in neural progenitor cells (NPCs) of *TRNP1* and the 22 TFs with enriched binding sites (motif weight ≥ 1) on the intron CREs. Each box represents the median, first and third quartiles with the whiskers indicating the furthest value no further than 1.5 * IQR from the box. Points indicate individual expression values. Vertical line indicates average expression across all 392 TFs (5.58), grey area: standard deviation (1.61). (**C**) Eight top enriched biological processes (Gene Ontology, Fisher's exact test p-value <0.05) of the 22 TFs. Background: all expressed TFs (392). (**D**) Variation in binding scores of the enriched TFs across catarrhines. Heatmaps indicate standardized binding scores (grey), gyrification index (GI) values (blue) and intron CRE activities (yellow) from the respective species. TF background colour indicates gene ontology assignment of the TFs to the two most significant biological processes. The bottom panel indicates the spatial position of the top binding site (motif score >3) for each TF on the human sequence. (**E**) Binding scores of three TFs (CTCF, ZBTB26, SOX8) are the best candidates to explain intron CRE activity, whereas only CTCF binding shows an association with the GI (phylogenetic generalized least squares [PGLS], likelihood ratio test [LRT] p-value <0.05). (**F**) Predicted intron CRE activity by the binding scores of the three TFs vs. the measured intron CRE activity across catarrhines.

The online version of this article includes the following figure supplement(s) for figure 4:

**Figure supplement 1.** TRNP1 expression in human and cynomolgus macaque (*Macaca fascicularis*) cell lines.

**Figure supplement 2.** Human genome tracks for the TRNP1 locus (hg19).

Of note, the effect of structural changes appears stronger than the effect of regulatory changes. This is contrary to the notion that regulatory changes should be the more likely targets of selection as they are more cell-type specific (*Carroll, 2008*) (but see also *Hoekstra and Coyne, 2007*). However, current measures of regulatory activity are inherently less precise than counting amino acid changes,

which will necessarily deflate the estimated association strength (*Danko et al., 2018*; *Berthelot et al., 2018*; *Huber et al., 2020*). Not only is gene regulation cell-type and time-dependent, but regulatory elements also evolve much faster, making a comprehensive and informative comparison across large phylogenies much more difficult. Moreover, while MPRAs function well in deciphering the regulatory activities of individual CREs, they are still limited in their in vivo interpretation. In any case, our analysis suggests that evolution likely combined both regulatory and structural evolution to modulate TRNP1 activity.

The MPRA also allowed to identify TFs that have a binding site enrichment to the intron CRE and are likely direct regulators of TRNP1. These include INSM1 (*Tavano et al., 2018*), which also has been shown to control NEC-to-neural-progenitor transition, as well as other relevant factors with increased activity in human neural stem and progenitor cells during early cortical development compared to later stages, such as TFAP2A, NFIC, TCF3, KLF12, and again INSM1 (*Trevino et al., 2021*; *de la Torre-Ubieta et al., 2018*). Among the enriched TFs that bind to the intron CRE, CTCF had the strongest association with gyrification. Although CTCF is best known for its insulating properties, it can also act as transcriptional activator and recruit co-factors in a lineage-specific manner (*Arzate-Mejía et al., 2018*). In neural progenitors, CTCF loss causes severe impairment in proliferative capacity through the increase in premature cell cycle exit, which results in drastically reduced progenitor pool and early differentiation (*Watson et al., 2014*). The overlapping molecular roles of TRNP1 and CTCF in neural progenitors support the possibility that TRNP1 is among the cell-fate determinants downstream of CTCF (*Wu et al., 2006*; *Delgado-Olguín et al., 2011*). Differences between species in CTCF binding strength and/or length to the intron CRE might have direct consequences for the binding of additional TFs, TRNP1 expression, and the resulting progenitor pool. However, the effects of CTCF binding in vitro and in vivo might differ and the exact mechanism, including the developmental timing and cellular context in which this might be relevant, is yet to be disentangled.

Independent from the mechanisms and independent whether caused by regulatory or structural changes, it is relevant how an increased TRNP1 activity could alter brain development. When overexpressing *Trnp1* in aRGCs of developing mice (E13) and ferrets (E30), aRGC proliferation increases (*Stahl et al., 2013*; *Pilz et al., 2013*; *Martínez-Martínez et al., 2016*). Similarly, overexpression of *Trnp1* increases proliferation in vitro in NSCs (*Stahl et al., 2013*; *Esgleas et al., 2020*) or breast cancer cells (*Volpe et al., 2006*). Hence, TRNP1 evolution could contribute to evolving a larger brain by increasing the pool of aRGCs. In addition, increases in brain size and especially increases in cortical folding are highly dependent on increases in proliferation of BPs, in particular bRGCs (*Pinson and Huttner, 2021*; *Del-Valle-Anton and Borrell, 2022*; *Villalba et al., 2021*). Remarkably, recent evidence indicates that *Trnp1* could be important also for the proliferation of BPs (*Kerimoglu et al., 2021*): Firstly, in contrast to non-proliferating BPs from mice, proliferating BPs from human do express TRNP1 (*Kerimoglu et al., 2021*). Furthermore, when activating expression of *Trnp1* using CRISPRa at E14.5, more proliferating BPs and induction of cortical folding is observed (*Kerimoglu et al., 2021*). Hence, a more active TRNP1 can increase proliferation in aRGCs and BPs and this could cause the observed co-evolution with brain size and cortical folding. *TRNP1* is the first case where analyses of protein sequence, regulatory activity, and protein activity across a larger phylogeny have been combined to investigate the role of a candidate gene in brain evolution. Functional evidence from evolutionary changes on the human lineage, for example, for ARHGAP11B and NOTCH2NL, but also phylogenetic evidence from correlating sequence changes with brain size changes (*Montgomery et al., 2016*; *Boddy et al., 2017*) indicate that a substantial number of genes could adapt their function when brain size changes in mammalian lineages. Improved genome assemblies (*Rhie et al., 2021*) will decisively improve phylogenetic approaches (*Cavassim et al., 2022*; *Stephan et al., 2022*; *Jourjine and Hoekstra, 2021*; *Smith et al., 2020*). In combination with the increased possibilities for functional assays due to DNA synthesis (*Chari and Church, 2017*) and comparative cellular resources across many species (*Enard, 2012*; *Housman and Gilad, 2020*; *Geuder et al., 2021*), this offers exciting possibilities to study the genetic basis of complex phenotypes within and across species.

## Materials and methods

### Sample collection and cell culture

#### Mouse strain and handling

Mouse handling and experimental procedures were performed in accordance with German and European Union guidelines and were approved by the State of upper Bavaria. All efforts were made to minimize suffering and number of animals. Two- to three-month female C57BL/6J wild-type mice were maintained in specific pathogen-free conditions in the animal facility, in 12:12 hr light/dark cycles and bred under standard housing conditions in the animal facility of the Helmholtz Center Munich and the Biomedical Center Munich. The day of the vaginal plug was considered E0.

#### Primary cerebral cortex harvesting and culture

E14 mouse (*M. musculus*) cerebral cortices were dissected, removing the ganglionic eminence, the olfactory bulb, the hippocampal anlage, and the meninges. Cells were mechanically dissociated with a fire polish Pasteur pipette. Cells were then seeded onto poly-D-lysine (PDL)-coated glass coverslips in DMEM-GlutaMAX (Dulbeccos's modified Eagles's medium) supplemented with 10% foetal calf serum (FCS) and 100 µg/mL Pen. Strep. and cultured at 37°C in a 5% $CO_2$ incubator.

#### Culture of HEK293T cells

HEK 293T cells (*H. sapiens*) were grown in DMEM supplemented with 10% FCS and 1% Pen. Strep. Cells were cultured in 10 cm flat-bottom dishes at 37°C in a 5% $CO_2$ environment and split every 2–3 days in a 1:10 ratio using 5 mL PBS to wash and 0.5 mL 0.25% Trypsin to detach the cells.

#### Culture of Neuro-2A cells

Neuro-2A cells (N2A) (ATCC; CCL-131, *M. musculus*) were cultured in Eagle's minimum essential medium (Thermo Fisher Scientific) with 10% FCS (Thermo Fisher Scientific) at 37°C in a 5% $CO_2$ incubator and split every 2–3 days in a 1:5 ratio using 5 mL PBS (Thermo Fisher Scientific) to wash and 0.5 mL 0.25% Trypsin (Thermo Fisher Scientific) to detach the cells.

#### Culture of neural progenitor cells

Neural progenitor cells of two human (*H. sapiens*) and one cynomolgus monkey (*M. fascicularis*) cell line (*Geuder et al., 2021*) were cultured at 37°C in a 5% $CO_2$ incubator on Geltrex (Thermo Fisher Scientific) in DMEM F12 (Fisher Scientific) supplemented with 2 mM GlutaMAX-I (Fisher Scientific), 20 ng/µL bFGF (Peprotech), 20 ng/µL hEGF (Miltenyi Biotec), 2% B-27 supplement (50×) minus vitamin A (Gibco), 1% N2 supplement 100× (Gibco), 200 µM L-ascorbic acid 2-phosphate (Sigma), and 100 µg/mL penicillin-streptomycin (Pen. Strep.) with medium change every second day. For passaging, NPCs were washed with PBS and then incubated with TrypLE Select (Thermo Fisher Scientific) for 5 min at 37°C. Culture medium was added and cells were centrifuged at 200 × *g* for 5 min. Supernatant was replaced by fresh culture medium and cells were transferred to a new Geltrex-coated dish. The cells were split every 2–3 days in a ratio of 1:3. All cell lines have been authenticated using RNA sequencing (RNA-seq), see *Geuder et al., 2021*, and the current study. Mycoplasma is regularly tested for using PCR-based test.

### Sequencing of *TRNP1* for primate species

#### Identification of CREs of *TRNP1*

DHS in the proximity to *TRNP1* (25 kb upstream, 3 kb downstream) were identified in human foetal brain and mouse embryonic brain DNase-seq datasets (*Vierstra et al., 2014*; *Bernstein et al., 2010*) downloaded from NCBI's Sequence Read Archive (see Appendix 1—key resources table ). Reads were mapped to human genome version hg19 and mouse genome version mm10 using NextGenMap with default parameters (NGM; version 0.0.1) (*Sedlazeck et al., 2013*). Peaks were identified with Hotspot version 4.0.0 using default parameters (*John et al., 2011*). Overlapping peaks were merged, and the union per species was taken as putative CREs of *TRNP1* (*Supplementary file 3a*). The orthologous regions of human *TRNP1* DNase peaks in 49 mammalian species were identified with reciprocal best hit using BLAT (v. 35x1) (*Kent, 2002*). Firstly, sequences of human *TRNP1* DNase peaks were extended

by 50 bases down- and upstream of the peak and the best matching sequence per peak region were identified with BLAT using the following settings: -t=DNA -q=DNA -stepSize=5 -repMatch=2253 -minScore=0 -minIdentity=0 -extendThroughN. These sequences were aligned back to hg19 using the same settings as above. The resulting best matching hits were considered reciprocal best hits if they fell into the original human *TRNP1* CREs. In total, 351 putative TRNP1 CRE sequences were identified, including human, mouse, and orthologous sequences.

## Cross-species primer design for sequencing

We sequenced TRNP1 coding sequences in six primates for which reference genome assemblies were either unavailable or very sparse and the ferret (*Mustela putorius furo*) where the sequence was incomplete (see *Supplementary file 1a*). For the missing primate sequences we used NCBI's tool Primer Blast (*Ye et al., 2012*) with the human *TRNP1* gene locus as a reference. Primer specificity was confirmed using the predicted templates in 12 other primate species available in Primer Blast. Following primers were used as they worked reliably in all six species (forward primer, GGGA GGAGTAAACACGAGCC; reverse primer, AGCCAGGTCATTCACAGTGG). For the ferret sequence, the genome sequence (MusPutFur1.0,) contained a gap in the TRNP1 coding sequence leading to a truncated protein. To recover the full sequence of TRNP1 we used the conserved sequence 5' of the gap and 3' of the gap as input for primer blast (primer sequences can be found in the analysis GitHub, see Data availability).

In order to obtain *TRNP1* CREs for the other primate species, we designed primers using primux (*Hysom et al., 2012*) based on the species with the best genome assemblies and subsequently tested them in closely related species in multiplexed PCRs. A detailed list of designed primer pairs per CRE and reference genome can be found in the analysis GitHub (see Data availability).

## Sequencing of target regions for primate species

Primate gDNAs were obtained from Deutsches Primaten Zentrum, DKFZ, and MPI Leipzig (see *Supplementary file 1b*). Depending on concentration, gDNAs were whole genome amplified prior to sequencing library preparation using GenomiPhi V2 Amplification Kit (Sigma). After amplification, gDNAs were cleaned up using SPRI beads (CleaNA). Both *TRNP1* coding regions and CREs were resequenced starting with a touchdown PCR to amplify the target region followed by a ligation and Nextera XT library construction. *TRNP1* coding regions were sequenced as 250 bases paired end with dual indexing on an Illumina MiSeq, the CRE libraries libraries were sequenced 50 bp paired end on an Illumina HiSeq 1500.

## Assembly of sequenced regions

Reads were demultiplexed using deML (*Renaud et al., 2015*). The resulting sequences per species were subsequently trimmed to remove PCR handles using cutadapt (version 1.6) (*Martin, 2011*). For sequence reconstruction, Trinity (version 2.0.6) in reference-guided mode was used (*Grabherr et al., 2011*). The reference here is defined as the mapping of sequences to the closest reference genome with NGM (version 0.0.1) (*Sedlazeck et al., 2013*). Furthermore, read normalization was enabled and a minimal contig length of 500 was set. The sequence identity of the assembled contigs was validated by BLAT (*Kent, 2002*) alignment to the closest reference *TRNP1* as well as to the human *TRNP1*. The assembled sequence with the highest similarity and expected length was selected per species.

The same strategy was applied to the resequenced ferret genomic sequence, except that we used bwa-mem2 (*Vasimuddin et al., 2019*) for mapping and for the assembly with Trinity we set minimal contig length to 300 (reference genome musFur1). Only the part covering the 3' end (specifically, the last 107 AAs) was successfully assembled, however, luckily, MusFur1 genome assembly already provides a good-quality assembly for the 5' end of the protein. The overlapping 36 AAs (108 nucleotides) between both sources had a 100% agreement on the nucleotide sequence level, hence we collapsed the sequences from both sources to yield a full-length protein-coding sequence. In a neighbour joining tree, where we included the nucleotide sequences from all 30 mammalian TRNP1 orthologues, ferret sequence was placed within the other carnivore sequences (between cat and a branch leading to seal, sea lion) as expected given the phylogenetic relationships of these species.

### *TRNP1* coding sequence retrieval and alignment

Human TRNP1 protein sequence was retrieved from UniProt database (*UniProt Consortium, 2019*) under accession number Q6NT89. We used the human TRNP1 in a tblastn (*Camacho et al., 2009*) search of genomes from 45 species, without any repeat masking specified in *Supplementary file 1a* (R-package rBLAST version 0.99.2). The resulting sequences were re-aligned with PRANK (*Löytynoja, 2021*) (version 150803), using the mammalian tree from *Bininda-Emonds et al., 2007*.

### Control gene set selection and alignment

Control genes were selected using consensus coding sequence (CCDS) dataset for human GRCh38.p12 genome (35,138 coding sequences, release 23) (*Pujar et al., 2018*). RBB (*Kent, 2002*) strategy was applied to identify the orthologous sequences in the other 29 species using -q=prot -t=dnax blat settings. We picked the best matching sequence per CDS in each species using a score based on the BLOSUM62 substitution matrix (*Henikoff and Henikoff, 1992*) and gapOpening = 3, gapExtension = 1 penalties, and requiring at least 30% of the human sequence to be found in the other species. This sequence was extracted and the same strategy was applied when blatting the orthologous sequence to the human genome. If the target sequence with the best score overlaps at least 10% of the original CDS positions, it was kept. To have a comparable gene set to TRNP1 in terms of statistical power and alignment quality, we selected all genes that had a similar human coding sequence length as TRNP1 (≥291 and ≤999 nucleotides) and 1 coding exon (322 out of the total of 1088 1-exon similar-length candidates prior to RBB). If RBB returned multiple matches per species per sequence with the same highest alignment score to the human sequence, we kept these only if the matching sequences were identical, which resulted in 274 genes. We further filtered for genes with all orthologous sequences of length at least 50% and below 200% relative to the length of the respective human protein-coding orthologue (257 genes). These were aligned using PRANK (*Löytynoja, 2021*) as for TRNP1, and manually inspected. One hundred and twelve alignments were optimal, and we could get additional 22 high-quality alignments by searching orthologues in additional genome versions using the previously described RBB strategy (gorilla gorGor5.fa, dolphin GCF_011762595.1_mTurTru1, wild boar GCF_000003025.6_Sscrofa11.1, rhesus macaque GCF_003339765.1_Mmul_10, olive baboon GCA_000264685.2_Panu_3.0) and redoing the alignment. Gene TREX1 turned out to have two CCDS included: CCDS2769.1, CCDS59451.1. As these are not independent, we randomly kept only one CCDS (CCDS2769.1). Alignment information content per protein-coding sequence (TRNP1 and 133 controls) was quantified as the average total branch length reduction across positions as a result of gaps using the following formula:

$$\overline{\lambda}_{red} = \frac{1}{p} \sum_{i=1}^{p} \frac{\lambda_i}{\lambda_t},$$

where $i$ to $p$ is alignment position, $\lambda_i$ is the total branch length at position $i$, $\lambda_t$ is the total branch length of the full 30 species tree. All branch lengths were taken from the pruned mammalian tree from *Bininda-Emonds et al., 2007*. This information per protein can be found in *Supplementary file 1f*, column AlnInfoContent.

## Evolutionary sequence analysis

For all evolutionary analyses, the pruned mammalian tree from *Bininda-Emonds et al., 2007*, was provided to the respective program.

### Estimation of the total tree length for *dS* and *dN/dS*

Program codeml from PAML software (*Yang, 1997*) (version 4.8) was used to obtain the total tree length for *dS* and *dN*. *dN/dS* was calculated as the ratio between the two parameters. Branch free-ratio model was ran on TRNP1 and 133 control protein-coding sequences using the following settings seqtype = 1, CodonFreq = 2, clock = 0, aaDist = 0, model = 1. We required the *log(dS)* tree length to be <3× SD away from the average, leading to the exclusion of one protein CCDS34575.1, resulting in a set containing 132 control sequence alignments and TRNP1.

## Inferring correlated evolution using Coevol

Coevol (*Lartillot and Poujol, 2011*) (version 1.4) was utilized to infer the covariance between TRNP1 and control protein evolutionary rate $\omega$ with three morphological traits (brain size, GI, and body mass) across species (*Supplementary file 1c*). Coevol is a Bayesian phylogenetic approach that jointly models substitution rates and continuous trait changes as a multivariate Brownian motion, yielding an estimate of the correlation structure between these variables, while reconstructing divergence times and ancestral traits. Simultaneous parameter estimation within the same framework helps avoiding error propagation.

For each model, the MCMC was run three times for at least 10,000 cycles, using the first 1000 as burn-in. For TRNP1 and 124 control proteins all parameters have a relative difference <0.3 and effective size >50, indicating good convergence, 8 control proteins did not reach convergence and were thereby excluded from further analyses. We report the average posterior probabilities (*pp*), the average marginal and partial correlations of the full model (*Supplementary file 1e*) and the separate models where including only either one of the three traits (*Supplementary file 1e*). The PP for a negative correlation are given by $1 - pp$. These were back-calculated to make them directly comparable, independently of the correlation direction, that is, higher $pp$ means more statistical support for the respective correlation.

## Identification of sites under positive selection

Program codeml from PAML software (*Yang, 1997*) (version 4.8) was used to infer whether a significant proportion of TRNP1 protein sites evolve under positive selection across the phylogeny of 45 species, setting seqtype = 1, CodonFreq = 2, clock = 0, aaDist = 0, model = 0. Site models M8 (NSsites = 8) and M7 (NSsites = 7) were compared (*Yang et al., 2000*), that allow $\omega$ to vary among sites across the phylogenetic tree, but not between branches. M7 and M8 are nested with M8 allowing for sites under positive selection with $\omega_s$. LRT with 2 degrees of freedom was used to compare these models. Naive empirical Bayes (NEB) analysis was used to identify the specific sites under positive selection ($Pr(\omega > 1) > 0.95$).

## Proliferation assay

### Plasmid construction

The five *TRNP1* orthologous sequences containing the restriction sites BamHI and XhoI were synthesized by GeneScript. All plasmids for expression were first cloned into a pENTR1a gateway plasmid described in *Stahl et al., 2013*, and then into a Gateway (Invitrogen) form of pCAG-GFP (kind gift of Paolo Malatesta). The gateway LR-reaction system was used to then sub-clone the different TRNP1 orthologues into the pCAG destination vectors.

### Primary cerebral cortex transfection

Primary cerebral cortex cultures were established as outlined under experimental model and subject details. Plasmids were transfected with Lipofectamine 2000 (Life Technologies) according to the manufacturer's instruction 2 hr after seeding the cells onto PDL-coated coverslips. One day later cells were washed with phosphate buffered saline (PBS) and then fixed in 4% paraformaldehyde (PFA) in PBS and processed for immunostaining.

### Immunostaining

Cells plated on PDL-coated glass coverslips were blocked with 2% BSA, 0.5% Triton-X (in PBS) for 1 hr prior to immunostaining. Primary antibodies (chicken alpha-GFP, Aves Labs: GFP-1010 and rabbit alpha-Ki67, abcam: ab92742) were applied in blocking solution overnight at 4°C. Fluorescent secondary antibodies were applied in blocking solution for 1 hr at room temperature. DAPI (4',6-diamidin-2-phenylindol, Sigma) was used to visualize nuclei. Stained cells were mounted in Aqua Polymount (Polysciences). All secondary antibodies were purchased from Life Technologies. Representative high-quality images were taken using an Olympus FV1000 confocal laser-scanning microscope using 20×/0.85 NA water immersion objective. Images used for quantification were taken using an epifluorescence microscope (Zeiss, Axio ImagerM2) equipped with a 20×/0.8 NA and 63×/1.25 NA

oil immersion objectives. Postimage processing with regard to brightness and contrast was carried out where appropriate to improve visualization, in a pairwise manner.

## Proliferation rate calculation using logistic regression

The proportion of successfully transfected cells that proliferate under each condition (Ki67-positive/GFP-positive) was modeled using logistic regression (R-package stats (version 4.0.3), glm function) with logit link function $logit(p) = log(\frac{p}{1-p})$, for $0 \leq p \leq 1$, where $p$ is the probability of success. The absolute number of GFP-positive cells were added as weights. Model selection was done using LRT within ANOVA function from stats. Adding the donor mouse as a batch improved the models (*Supplementary file 2a*).

To back-calculate the absolute proliferation probability (i.e., rate) under each condition, intercept of the respective model was set to zero and the inverse logit function $\frac{e^{\beta_i X_i}}{1+e^{\beta_i X_i}}$ was used, where $i$ indicates condition (*Supplementary file 2b*). Two-sided multiple comparisons of means between the conditions of interest were performed using glht function (Tukey test, user-defined contrasts) from R package multcomp (version 1.4-13) (*Supplementary file 2c*).

## Phylogenetic modeling of proliferation rates using generalized least squares

The association between the induced proliferation rates for each TRNP1 orthologue and the brain size or GI of the respective species was analysed using generalized least squares (R-package nlme, version 3.1-143), while correcting for the expected correlation structure due to phylogenetic relation between the species. The expected correlation matrix for the continuous trait was generated using a Brownian motion (*Felsenstein, 1985*; *Martins and Hansen, 1997*) (ape [version 5.4], using function corBrownian). The full model was compared to a null model using the LRT. Residual $R^2$ values were calculated using R2.resid function from R package RR2 (version 1.0.2).

## **Massively parallel reporter assay**

### MPRA library design

A total of 351 potential *TRNP1* CRE sequences were identified as outlined before. Based on these, the MPRA oligos were designed as 94mers, where larger sequences were covered by sliding window by 40 bases, resulting in 4950 oligonucleotide sequences, that are flanked by upstream and downstream priming sites and KpnI/XbaI restriction cut sites as in the original publication (*Melnikov et al., 2012*). Barcode tag sequences were designed so that they contain all four nucleotides at least once, do not contain stretches of four identical nucleotides, do not contain microRNA seed sequences (retrieved from microRNA Bioconductor R package, version 1.28.0), and do not contain restriction cut site sequences for KpnI nor XbaI. The full library of designed oligonucleotides can be found on GitHub (see Data availability).

### MPRA library construction

We modified the original MPRA protocol (*Melnikov et al., 2012*) by using a lentiviral delivery system as previously described (*Inoue et al., 2017*), introducing GFP instead of nanoluciferase and changing the sequencing library preparation strategy. In brief, oligonucleotide sequences (Custom Array) were amplified using emulsion PCR (Micellula Kit, roboklon) and introduced into the pMPRA plasmid as described previously. The nanoluciferase sequence used in the original publication was replaced by EGFP using Gibson cloning and subsequent insertion into the enhancer library using restriction enzyme digest as in the original publication. Using SFiI the assembled library was transferred into a suitable lentiviral vector (pMPRAlenti1, Addgene #61600).

Primer sequences and plasmids used in the MPRA can be found in the analysis GitHub (see Data availability). To ensure maximum library complexity, transformations that involved the CRE library were performed using electroporation (NEB 10-beta electrocompetent *Escherichia coli*), in all other cloning steps chemically competent *E. coli* (NEB 5-alpha) were used.

Lentiviral particles were produced according to standard methods in HEK 293T cells (*Dull et al., 1998*). The MPRA library was co-transfected with third generation lentiviral plasmids (pMDLg/pRRE, pRSV-Rev, pMD2.G; Addgene #12251, #12253, #12259) using Lipofectamine 3000. The lentiviral particle containing supernatant was harvested 48 hr post transfection and filtered using 0.45 µm PES

syringe filters. Viral titer was determined by infecting N2A cells (ATCC CCL-131) and counting GFP-positive cells. To this end, N2A cells were infected with a 50/50 volume ratio of viral supernatant to cell suspension with addition of 8 µg/mL Polybrene. Cells were exposed to the lentiviral particles for 24 hr until medium was exchanged. Selection was performed using blasticidin starting 48 hr after infection.

## MPRA lentiviral transduction

The transduction of the MPRA library was performed in triplicates on two *H. sapiens* and one *M. fascicularis* NPC lines generated as described previously (*Geuder et al., 2021*). 2.5 × 10⁵ NPCs per line and replicate were dissociated, dissolved in 500 µL cell culture medium containing 8 µg/mL Poly-brene and incubated with virus at MOI 12.7 for 1 hr at 37°C in suspension (*Nakai et al., 2018*). There-after, cells were seeded on Geltrex and cultured as described above. Virus containing medium was replaced the next day and cells were cultured for additional 24 hr. Cells were collected, lysed in 100 µL TRI reagent, and frozen at –80°C.

## MPRA sequencing library generation

As input control for RNA expression, DNA amplicon libraries were constructed using 100–500 pg plasmid DNA. Library preparation was performed in two successive PCRs. A first PCR introduced the 5′ transposase mosaic end using overhang primers, this was used in the second PCR (Index PCR) to add a library-specific index sequence and Illumina Flow Cell adapters. The Adapter PCR was performed in triplicates using DreamTaq polymerase (Thermo Fisher Scientific). Subsequently 1–5 ng of the Adapter PCR product were subjected to the Index PCR using Q5 polymerase.

Total RNA from NPCs was extracted using the Direct-zol RNA Microprep Kit (Zymo Research). Five hundred ng of RNA were subjected to reverse transcription using Maxima H Minus RT (Thermo Fisher Scientific) with oligo-dT primers. Fifty ng of cDNA were used for library preparation and processed as described for plasmid DNA.

Plasmid and cDNA libraries were pooled and quality was evaluated using capillary gel electropho-resis (Agilent Bioanalyzer 2100). Sequencing was performed on an Illumina HiSeq 1500 instrument using a single-index, 50 bp, paired-end protocol.

## MPRA data processing and analysis

MPRA reads were demultiplexed with deML (*Renaud et al., 2015*) using i5 and i7 adapter indices from Illumina. Next, we removed barcodes with low sequence quality, requiring a minimum Phred quality score of 10 for all bases of the barcode (zUMIs, fqfilter.pl script; *Parekh et al., 2018*). Further-more, we removed reads that had mismatches to the constant region (the first 20 bases of the GFP sequence TCTAGAGTCGCGGCCTTACT). The remaining reads that matched one of the known CRE-tile barcodes were tallied up resulting in a count table. Next, we filtered out CRE tiles that had been detected in only one of the three input plasmid library replicates (4202/4950). Counts per million were calculated per CRE tile per library (median counts: ~900k range: 590–1050k). Macaque replicate 3 was excluded due to its unusually low correlation with the other samples (Pearson's *r*). The final regulatory activity for each CRE tile per cell line was calculated as:

$$a_i = \frac{median(CPM_i)}{median(CPM_i)_p},$$ (1)

where *a* is regulatory activity, *i* indicates CRE tile, and *p* is the input plasmid library. Median was calcu-lated across the replicates from each cell line.

Given that each tile was overlapping with two other tiles upstream and two downstream, we calcu-lated the total regulatory activity per CRE region in a coverage-sensitive manner, that is, for each posi-tion in the original sequence, mean per-bp-activity across the detected tiles covering it was calculated. The final CRE region activity is the sum across all base positions.

$$a_r = \sum_{b=1}^{k} \frac{1}{n} \sum_{i=1}^{n} \frac{a_i}{l_i},$$ (2)

where $a_r$ is regulatory activity of CRE region *r*, $b = 1, ..., k$ is the base position of region *r*, $i, ..., n$ are tiles overlapping the position *b*, $a_i$ is tile activity from *Equation 1* and $l_i$ is tile length. CRE activity and brain

phenotypes were associated with one another using PGLS analysis (see above). The number of species varied for each phenotype-CRE pair (brain size: min. 37 for exon 1, max. 48 for intron and downstream regions; GI: min. 32 for exon2, max. 37 for intron), therefore the activity of each of the seven CRE regions was used separately to predict either GI or brain size of the respective species.

## TF analysis

### RNA-seq library generation

RNA-seq was performed using the prime-seq method (*Janjic et al., 2022*). The full prime-seq protocol including primer sequences can be found at protocols.io (https://www.protocols.io/view/prime-seq-s9veh66). Here, we used 10 ng of the isolated RNA from the MPRA experiment and subjected it to the prime-seq protocol. Sequencing was performed on an Illumina HiSeq 1500 instrument with the following setup: read 1 16 bases, read 2 50 bases, and i7 index read 8 bases.

### RNA-seq data processing

Bulk RNA-seq data was generated from the same nine samples (three cell lines, three biological replicates each) that were assayed in the MPRA. Raw read fastq files were pre-processed using zUMIs (version 2.4.5b) (*Parekh et al., 2018*) together with STAR (version STAR_2.6.1c) (*Dobin et al., 2013*) to generate expression count tables for barcoded UMI data. Reads were mapped to human reference genome (hg38, Ensembl annotation GRCh38.84). Further filtering was applied keeping genes that were detected in at least 7/9 samples and had on average more than 7 counts, resulting in 17,306 genes. For further analysis, we used normalized and variance stabilized expression estimates as provided by DESeq2 (*Love et al., 2014*), using a model ~0+ clone. Differential expression testing between clone pairs was carried out using Benjamini and Hochberg-corrected Wald test as implemented in DESeq2.

### TFBS motif analysis on the intron CRE sequence

TF position frequency matrices were retrieved from JASPAR CORE 2020 (*Fornes et al., 2020*), including only non-redundant vertebrate motifs (746 in total). These were filtered for the expression in our NPC RNA-seq data, leaving 392 TFs with 462 motifs in total.

A hidden Markov model-based program Cluster-Buster (*Frith et al., 2003*) (compiled on 13 June 2019) was used to infer the enriched TF binding motifs on the intron sequence. Firstly, the auxiliary program Cluster-Trainer was used to find the optimal gap parameter between motifs of the same cluster and to obtain weights for each TF based on their motif abundance per kb across catharrine intron CREs from 10 species with available GI measurements. Weights for each motif suggested by Cluster-Trainer were supplied to Cluster-Buster that we used to find clusters of regulatory binding sites and to infer the enrichment score for each motif on each intron sequence. The program was run with the following parameters: –g3 –c5 –m3.

To identify the most likely regulators of *TRNP1* that bind to its intron sequence and might influence the evolution of gyrification, we filtered for the motifs that were most abundant across the intron sequences (Cluster-Trainer weights >1). These motifs were distinct from one another (mean pairwise distance 0.72). Gene set enrichment analysis contrasting the TFs with the highest binding potential with the other expressed TFs was conducted using the Bioconductor package topGO (*Alexa, 2009*) (version 2.40.0) (*Supplementary file 3*), setting the following parameters: ontology='BP', nodeSize = 20, algorithm = 'elim', statistic = 'fisher'. PGLS model was applied as previously described, using Cluster-Buster binding scores across catharrine intron CRE sequences as predictors and predicting either intron activity or GI from the respective species. The relevance of the three TFs that were associated with intron activity was then tested using an additive model and comparing the model likelihoods with reduced models where either of these were dropped.

### Retrieving public data

Annotations and coordinates of enhancers showing gained activity in humans based on H3K27ac and H3K4me2 histone marks were downloaded from GSE63648 (*Reilly et al., 2015*) as bed files from the section Supplementary files.

CTCF ChiP-seq data from human neural progenitor cells (line H9) was retrieved from ENCODE (*Encode Project Consortium, 2012*) (doi:10.17989/ENCSR125NBL). All samples were consistent regarding TRNP1 CTCF ChIP-seq landscape. We depict read distribution using BigWig file of sample ENCFF896TQG.

Human Hi-C data (*Won et al., 2016*) on TAD positions in germinal zone at week 8 was retrieved as a coordinate file in bed format using GEO accession GSE77565.

## Quantification and statistical analysis

Data visualizations and statistical analysis was performed using R (version 4.0) (*R Development Core Team, 2019*). Details of the statistical tests performed in this study can be found in the main text as well as the Materials and methods section and *Supplementary files 1–3*. For display items all relevant parameters like sample size (*n*), type of statistical test, significance thresholds, degrees of freedom, as well as standard deviations can be found in the figure legends.

## Resource availability

### Lead contact

Further information and requests for resources and reagents should be directed to and will be fulfilled by the lead contact, Wolfgang Enard (enard@bio.lmu.de).

### Materials availability

Plasmids and cell lines used in this work will be available upon request.

## Acknowledgements

This work was supported by the Deutsche Forschungsgemeinschaft (DFG) through LMUexcellent, SFB1243 (Subproject A14/A15 to WE and IH, respectively), DFG grant HE 7669/1-1 (to IH) and the advanced ERC grants ChroNeuroRepair and NeuroCentro (to MG) and the Cyliax foundation (to WE). We want to thank Christian Roos from the German Primate Center for providing genomic DNA from primates, Deeksha for providing Ferret fibroblast cells, project students Gunnar Kuut and Fatih Sarigoel for helping to generate TRNP1 orthologous sequences, Nikola Vuković for helping to establish the MPRA assay, Nika Foglar and Reza Rifat for helping with the proliferation assays and Christoph Neumayr, Tamina Dietl for helping in data analysis.

## Additional information

### Funding

| Funder | Grant reference number | Author |
| --- | --- | --- |
| Cyliax Foundation | | Wolfgang Enard |
| Deutsche Forschungsgemeinschaft | 458247426 | Wolfgang Enard Ines Hellmann |
| Deutsche Forschungsgemeinschaft | 407541155 | Ines Hellmann |
| European Research Council | ChroNeuroRepair | Magdelena Götz |
| European Research Council | NeuroCentro | Magdelena Götz |

The funders had no role in study design, data collection and interpretation, or the decision to submit the work for publication.

### Author contributions

Zane Kliesmete, Data curation, Software, Formal analysis, Validation, Investigation, Visualization, Methodology, Writing – original draft, Writing – review and editing, Collected, integrated and analysed all

data; Lucas Esteban Wange, Validation, Investigation, Methodology, Writing – original draft, Writing – review and editing, Conducted the MPRA assay; Beate Vieth, Data curation, Methodology, Designed all initial sequence acquisitions; Miriam Esgleas, Validation, Investigation, Methodology, Designed and conducted the proliferation assay; Jessica Radmer, Validation, Investigation, Methodology, Primate cell culture work and MPRA conduction; Matthias Hülsmann, Investigation, Methodology, MPRA conduction; Johanna Geuder, Investigation, Methodology, Primate cell culture work; Daniel Richter, Methodology, MPRA conduction; Mari Ohnuki, Methodology, Primate cell culture work; Magdelena Götz, Conceptualization, Resources, Supervision, Funding acquisition, Project administration, Writing – review and editing, Proposed the project; Ines Hellmann, Conceptualization, Resources, Software, Supervision, Funding acquisition, Writing – original draft, Project administration, Writing – review and editing; Wolfgang Enard, Conceptualization, Resources, Supervision, Funding acquisition, Writing – original draft, Project administration, Writing – review and editing

**Author ORCIDs**
Ines Hellmann ⓘD http://orcid.org/0000-0003-0588-1313
Wolfgang Enard ⓘD http://orcid.org/0000-0002-4056-0550

**Decision letter and Author response**
Decision letter https://doi.org/10.7554/eLife.83593.sa1
Author response https://doi.org/10.7554/eLife.83593.sa2

## Additional files

### Supplementary files

• Supplementary file 1. Summaries of all information for the Coevol analyses, including the data sources for genome sequence and phenotype information as well as relevant Coevol outputs. Source information on TRNP1 protein sequences (1a), primate gDNA (1b), phenotype information (1c) as well as detailed results from PAML (*Yang, 1997*) (1d) and Coevol (*Lartillot and Poujol, 2011*) results for TRNP1 and the control proteins (1f, 1e, 1g).

• Supplementary file 2. Model selection for NSC proliferation (2a) as well as proliferation rates based on the selected model (2b) and statistical testing of pairwise differences (2c).

• Supplementary file 3. Analyses of TRNP1 CREs and their activities and a characterization of TF binding sites within. TRNP1 DNase hypersensitive sites (3a), phylogenetic generalized least squares (PGLS) model selections using likelihood ratio test for all seven CREs and the whole phylogeny (3b) as well as only the intron CRE in Old World monkeys and great apes (3c) and enriched gene ontologies based on the transcription factors (TFs) with binding site enrichment in the intron CRE (3d).

• MDAR checklist

### Data availability

The RNA-seq data used in this manuscript have been submitted to Array Express (https://www.ebi.ac.uk/arrayexpress/) under the accession number E-MTAB-9951. The MPRA data have been submitted to Array Express under accession number E-MTAB-9952. Additional primate sequences for TRNP1 have been submitted to GenBank (https://www.ncbi.nlm.nih.gov/genbank/) under the accession numbers MW373535–MW373709, and the ferret sequence under the accession number OP484343. A compendium containing processing scripts and detailed instructions to reproduce the analysis, as well as the most relevant data tables from this manuscript are available on the following GitHub repository: https://github.com/Hellmann-Lab/Co-evolution-TRNP1-and-GI (copy archived at *Kliesmete, 2023*).

The following datasets were generated:

| Author(s) | Year | Dataset title | Dataset URL | Database and Identifier |
|---|---|---|---|---|
| Kliesmete Z, Wange LE, Vieth B, Esgleas M, Radmer J, Hülsmann M, Geuder J, Richter D, Ohnuki M, Götz M, Hellmann I, Enard W | 2021 | RNA-seq of two human and one cynomologous NPC line to assay activity of DNAse1 hypersensitive sites in the proximity of the Trnp1 gene | https://www.ebi.ac.uk/arrayexpress/experiments/E-MTAB-9951/ | ArrayExpress, E-MTAB-9951 |
| Kliesmete Z, Wange LE, Vieth B, Esgleas M, Radmer J, Hülsmann M, Geuder J, Richter D, Ohnuki M, Götz M, Hellmann I, Enard W | 2021 | MPRA of two human and one cynomologous NPC line to assay activity of DNAse1 hypersensitive sites in the proximity of the Trnp1 gene | https://www.ebi.ac.uk/arrayexpress/experiments/E-MTAB-9952/ | ArrayExpress, E-MTAB-9952 |
| Kliesmete Z, Wange LE, Vieth B, Esgleas M, Radmer J, Huelsmann M, Geuder J, Richter D, Ohnuki M, Hellmann I, Enard W | 2021 | *Homo sapiens* TMF-regulated nuclear protein 1 (TRNP1) gene, complete cds | https://www.ncbi.nlm.nih.gov/nuccore/MW373535 | NCBI Nucleotide, MW373535 |
| Kliesmete Z, Wange LE, Vieth B, Esgleas M, Radmer J, Huelsmann M, Geuder J, Richter D, Ohnuki M, Hellmann I, Enard W | 2021 | Chlorocebus aethiops TMF-regulated nuclear protein 1 (TRNP1) gene, complete cds | https://www.ncbi.nlm.nih.gov/nuccore/MW373536 | NCBI Nucleotide, MW373536 |
| Kliesmete Z, Wange LE, Vieth B, Esgleas M, Radmer J, Huelsmann M, Geuder J, Richter D, Ohnuki M, Goetz M, Hellmann I, Enard W | 2021 | Cercopithecus mitis TMF-regulated nuclear protein 1 (TRNP1) gene, partial cds | https://www.ncbi.nlm.nih.gov/nuccore/MW373537 | NCBI Nucleotide, MW373537 |
| Kliesmete Z, Wange LE, Vieth B, Esgleas M, Radmer J, Huelsmann M, Geuder J, Richter D, Ohnuki M, Hellmann I, Enard W | 2021 | Papio anubis TMF-regulated nuclear protein 1 (TRNP1) gene, complete cds | https://www.ncbi.nlm.nih.gov/nuccore/MW373538 | NCBI Nucleotide, MW373538 |
| Kliesmete Z, Wange LE, Vieth B, Esgleas M, Radmer J, Huelsmann M, Geuder J, Richter D, Ohnuki M, Goetz M, Hellmann I, Enard W | 2021 | Mandrillus sphinx TMF-regulated nuclear protein 1 (TRNP1) gene, complete cds | https://www.ncbi.nlm.nih.gov/nuccore/MW373539 | NCBI Nucleotide, MW373539 |
| Kliesmete Z, Wange LE, Vieth B, Esgleas M, Radmer J, Huelsmann M, Geuder J, Richter D, Ohnuki M, Goetz M, Hellmann I, Enard W | 2021 | Macaca leonina TMF-regulated nuclear protein 1 (TRNP1) gene, partial cds | https://www.ncbi.nlm.nih.gov/nuccore/MW373540 | NCBI Nucleotide, MW373540 |
| Kliesmete Z, Wange LE, Vieth B, Esgleas M, Radmer J, Huelsmann M, Geuder J, Richter D, Ohnuki M, Goetz M, Hellmann I, Enard w | 2022 | Mustela putorius TMF-regulated nuclear protein 1 (TRNP1) gene, partial cds | https://www.uniprot.org/uniprotkb/Q80ZI1/entry/OP484343 | UniProt, OP484343 |

The following previously published datasets were used:

| Author(s) | Year | Dataset title | Dataset URL | Database and Identifier |
|---|---|---|---|---|
| Vierstra J, Rynes E, Sandstrom R, Thurman RE, Zhang M, Canfield T, Sabo PJ, Byron R, Hansen RS, Johnson AK, Vong S, Lee K, Bates D, Neri F, Diegel M, Giste E, Haugen E, Dunn D, Humbert R, Wilken MS, Josefowicz S, Samstein R, Chang K, Levassuer D, Disteche C, De Bruijn M, Rey TA, Skoultchi A, Rudensky A, Orkin SH, Papayannopoulou T, Treuting P, Selleri L, Kaul R, Bender MA, Groudine M, Stamatoyannopoulos JA | 2014 | Mouse regulatory DNA landscapes reveal global principles of cis-regulatory evolution | https://www.ncbi.nlm.nih.gov/geo/query/acc.cgi?acc=GSE51336 | NCBI Gene Expression Omnibus, GSE51336 |
| Stamatoyannopoulos JA | 2014 | Conservation of mouse-human trans-regulatory circuitry despite high cis-regulatory divergence | https://www.ncbi.nlm.nih.gov/geo/query/acc.cgi?acc=GSE51341 | NCBI Gene Expression Omnibus, GSE51341 |

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

# Appendix 1

## Appendix 1—key resources table

| Reagent type (species) or resource | Designation | Source or reference | Identifiers | Additional information |
|---|---|---|---|---|
| Gene (45 mammal species) | TRNP1 | See *Supplementary file 1a* | See *Supplementary file 1a* | See *Supplementary file 1a* |
| Strain, strain background (*E. coli*) | NEB 10-beta | New England Biolabs; Rowley, MA, United States | Cat# C3020K | Electrocompetent *E. coli* |
| Strain, strain background (*E. coli*) | NEB 5-alpha High Efficiency | New England Biolabs; Rowley, MA, United States | Cat# C2987I | Chemically competent *E. coli* |
| Cell line (*Macaca fascicularis*) | Cynomolgus Macaque NPC | This paper, based on *Geuder et al., 2021* | N15_39B2 | Macaca fascicularis neural progenitor cells |
| Cell line (*Mus musculus*) | N2A | ATCC; Manassas, VA, United States | CCL-131 | |
| Cell line (*Homo sapiens*) | HEK293T | ATCC; Manassas, VA, United States | CRL-11268 | |
| Cell line (*Homo sapiens*, female) | Human NPC 1 | This paper, based on *Geuder et al., 2021* | N4_29B5 | Human neural progenitor cells |
| Cell line (*Homo sapiens*, male) | Human NPC 2 | This paper, based on *Geuder et al., 2021* | N4_12 C2 | Human neural progenitor cells |
| Biological sample (*Mus musculus*) | Primary murine cerebral cortex cells (NSC) | This paper, based on *Esgleas et al., 2020* | primary | See Methods |
| Sequence-based reagent | MPRA oligo Library Trnp1 CRE | Custom Array; Redmond, WA, United States | custo | See https://github.com/Hellmann-Lab/Co-evolution-TRNP1-and-GI |
| Transfected construct (multiple species) | MPRA Library in lentiviral particles | This paper | custom | Lentiviral particles with pMPRA-lenti and TRNP1 CRE library |
| Antibody | rabbit anti Ki67 (monoclonal) | Abcam; Waltham, MA, United States | Cat# ab92742, Clone EPR3610 | 1:100 |
| Antibody | chicken anti-GFP (polyclonal) | Aves Labs; Davis, CA, United States | RRID: AB_2307313, Cat# GFP-1010, Polyclonal | 1:500 |
| Recombinant DNA reagent | pCAG-GFP_Gateway plasmid | Dr. Paolo Malatesta | NA | Kind gift of Dr. Paolo Malatesta |
| Recombinant DNA reagent | pMDLg/pRRE plasmid | Addgene; Waterton, MA, United States | Addgene 12251 | |
| Recombinant DNA reagent | pRSV-Rev plasmid | Addgene; Waterton, MA, United States | Addgene 12253 | |
| Recombinant DNA reagent | pMD2.G plasmid | Addgene; Waterton, MA, United States | Addgene 12259 | |
| Recombinant DNA reagent | pMPRAlenti1 plasmid | Addgene; Waterton, MA, United States | Addgene 61600 | Kind gift of Dr. Davide Cacchiarelli |
| Recombinant DNA reagent | pNL3.1[Nluc/minP] plasmid, SfiI restriction site mutated | Dr. Davide Cacchiarelli | NA | Kind gift of Dr. Davide Cacchiarelli |
| Recombinant DNA reagent | pMPRA1 plasmid | Addgene; Waterton, MA, United States | Addgene 49349 | Kind gift of Dr. Davide Cacchiarelli |
| Recombinant DNA reagent | pENTR1a plasmid | *Stahl et al., 2013* | pENTR1a | |
| Peptide, recombinant protein | hEGF | Miltenyi Biotec; Bergisch Gladbach, Germany | Cat#130-093-825 | |
| Peptide, recombinant protein | B-27 Supplement | Thermo Fisher Scientific; Waltham, MA, United States | Cat#12587–010 | |
| Peptide, recombinant protein | N2 Supplement | Thermo Fisher Scientific; Waltham, MA, United States | Cat#17502048 | |

*Appendix 1 Continued on next page*

*Appendix 1 Continued*

| Reagent type (species) or resource | Designation | Source or reference | Identifiers | Additional information |
|---|---|---|---|---|
| Peptide, recombinant protein | L-Ascorbic acid 2-phosphate | Sigma/Merck; St. Louis, MO, United States | Cat#A8960-5G | |
| Peptide, recombinant protein | poly-D-lysine | Sigma/Merck; St. Louis, MO, United States | Cat# A-003-E | |
| Peptide, recombinant protein | bFGF | PeproTech, Cranbury, New Jersey, United States | Cat#100-18B | |
| Commercial assay or kit | GenomiPhi V2 DNA-Amplification Kit | Sigma/Merck; St. Louis, MO, United States | Cat# GE25-6600-32 | |
| Commercial assay or kit | Gateway LR Clonase Enzyme mix | Thermo Fisher Scientific; Waltham, MA, United States | Cat# 11791019 | |
| Commercial assay or kit | Lipofectamine 2000 | Thermo Fisher Scientific; Waltham, MA, United States | Cat# 11668019 | |
| Commercial assay or kit | Lipofectamine 3000 | Thermo Fisher Scientific; Waltham, MA, United States | Cat# L3000015 | |
| Commercial assay or kit | Micellula DNA Emulsion & Purification Kit | Roboklon; Berlin, Germany | Cat# E3600-01 | |
| Commercial assay or kit | Agilent High Sensitivity DNA Kit | Agilent; Santa Clara, CA, United States | Cat# 5067–4626 | |
| Commercial assay or kit | Nextera XT DNA Library Preparation Kit | Illumina; San Diego, CA, United States | Cat# FC-131–1024 | |
| Chemical compound, drug | GlutaMax-I | Thermo Fisher Scientific; Waltham, MA, United States | Cat# 35050038 | |
| Chemical compound, drug | Blasticidin S HCl | Thermo Fisher Scientific; Waltham, MA, United States | Cat# R21001 | |
| Chemical compound, drug | DMEM-GlutaMAX | Thermo Fisher Scientific; Waltham, MA, United States | Cat# 10566016 | |
| Chemical compound, drug | Polybrene | Sigma/Merck; St. Louis, MO, United States | Cat# TR-1003-G | |
| Chemical compound, drug | TRI reagent | Sigma/Merck; St. Louis, MO, United States | Cat# T9424-200ML | |
| Chemical compound, drug | Geltrex | Thermo Fisher Scientific; Waltham, MA, United States | Cat# A1413302 | |
| Sequence-based reagent | Trnp1 CRE resequencing primers | Integrated DNA Technologies, Coralville, IO, United States | custom | See https://github.com/Hellmann-Lab/Co-evolution-TRNP1-and-GI |
| Sequence-based reagent | Trnp1 coding resequencing forward primer | Integrated DNA Technologies, Coralville, IO, United States | custom | GGGAGGAGTAAACACGAGCC |
| Sequence-based reagent | Trnp1 coding resequencing reverse primer | Integrated DNA Technologies, Coralville, IO, United States | custom | AGCCAGGTCATTCACAGTGG |
| Software, algorithm | Hotspot version 4.0.0 | *John et al., 2011*, http://www.uwencode.org/software/hotspot | NA | |
| Software, algorithm | BLAT version 35x1 | *Kent, 2002*, https://github.com/djhshih/blat | NA | |
| Software, algorithm | PriMux, compiled on 20 July 2014 | *Hysom et al., 2012*, https://sourceforge.net/projects/primux/ | NA | |
| Software, algorithm | deML version 1.1.3 | *Renaud et al., 2015*, https://github.com/grenaud/deml | NA | |
| Software, algorithm | cutadapt version 1.6 | *Martin, 2011*, https://anaconda.org/bioconda/cutadapt | NA | |

*Appendix 1 Continued on next page*

*Appendix 1 Continued*

| Reagent type (species) or resource | Designation | Source or reference | Identifiers | Additional information |
|---|---|---|---|---|
| Software, algorithm | Trinity version 2.0.6 | *Grabherr et al., 2011*, https://github.com/trinityrnaseq/trinityrnaseq/releases | NA | |
| Software, algorithm | rBLAST version 0.99.2 | https://github.com/mhahsler/rBLAST | NA | |
| Software, algorithm | PRANK version 150803 | *Löytynoja, 2021*, http://wasabiapp.org/software/prank/ | NA | |
| Software, algorithm | PAML version 4.8 | *Yang, 1997*, http://abacus.gene.ucl.ac.uk/software/paml.html | NA | |
| Software, algorithm | Coevol version 1.4 | *Lartillot and Poujol, 2011*, https://megasun.bch.umontreal.ca/People/lartillot/www/downloadcoevol.html | NA | |
| Software, algorithm | NextGenMap (NGM) version 0.0.1 | *Sedlazeck et al., 2013*, http://cibiv.github.io/NextGenMap/ | NA | |
| Software, algorithm | Primer Blast | *Ye et al., 2012* | NA | |
| Software, algorithm | zUMIs version 2.4.5b | *Parekh et al., 2018*, https://github.com/sdparekh/zUMIs | NA | |
| Software, algorithm | STAR version STAR_2.6.1 c | *Dobin et al., 2013*, https://github.com/alexdobin/STAR | NA | |
| Software, algorithm | DESeq2 version 1.26.0 | *Love et al., 2014*, Bioconductor | NA | |
| Software, algorithm | Cluster Buster, compiled on Jun 13 2019 | *Frith et al., 2003*, http://cagt.bu.edu/page/ClusterBuster_download | NA | |
| Software, algorithm | R version 3.6/4 | https://www.r-project.org/ | NA | |
| Software, algorithm | nlme version 3.1–143 | https://cran.r-project.org/web/packages/nlme/index.html | NA | |
| Software, algorithm | topGO version 2.40.0 | *Alexa, 2009*, https://bioconductor.org/packages/release/bioc/html/topGO.html | NA | |
| Software, algorithm | ape version 5.4 | https://cran.r-project.org/web/packages/ape/index.html | NA | |
| Software, algorithm | multcomp version 1.4–13 | https://cran.r-project.org/web/packages/multcomp/index.html | NA | |
| Software, algorithm | RR2 version 1.0.2 | https://cran.r-project.org/web/packages/rr2/index.html | NA | |

