## [Editor Report]

This is an important paper that combines comparative analysis and experimental assays to investigate the role of protein-coding and regulatory changes at TRNP1 in mammalian brain evolution. The evidence supporting a contribution of TRNP1 is convincing, although the strength of the link between protein-coding changes and trait evolution is stronger and more readily interpretable than the data on gene regulation. The work will be of interest to researchers interested in mammalian evolution, brain evolution, and evolutionary genetics.

---

## [Decision Letter]

**Decision letter after peer review:**

Thank you for submitting your article "Regulatory and coding sequences of TRNP1 co-evolve with brain size and cortical folding in mammals" for consideration by *eLife*. Your article has been reviewed by 3 peer reviewers, one of whom is a member of our Board of Reviewing Editors, and the evaluation has been overseen by Detlef Weigel as the Senior Editor. The reviewers have opted to remain anonymous.

Essential revisions:

1) The reviewers appreciated the effort to investigate both coding and regulatory changes and feel the MPRA study was well-designed. However, because the results of the MPRA are somewhat unclear (and necessarily incomplete as a test of regulatory function), please temper your conclusions accordingly. Additionally, clarify how you dealt with the multiple testing burden.

2) Please address the outstanding questions about the appropriateness of the background set of genes used for comparison of brain phenotype-evolutionary rate correlations. This could take the form of additional analysis, or minimally a discussion of the potential limitations and/or biases of the set you used.

*Reviewer #1 (Recommendations for the authors):*

I think this paper makes an important contribution to the literature. Genomic analyses that control for phylogenetic context/non-independence remain rare (at least in primates), and the integration of functional genomic analysis and experimentation is an important strength. I have a few concerns, however:

1. I am unclear on your interpretation of the correlation between brain size/gyrification and the rate of protein evolution-specifically, why a positive correlation should necessarily be expected. While comparative analyses often analyze correlated evolution between two traits (e.g., brain size and diet), this analysis seems more like analyzing the correlation between brain size and lineage-specific change in diet. If we assume that TRNP1 is a major driver of brain size, and selection on brain size fully accounts for selection on the gene, then the big-brained descendants of a big-brained ancestor might show strongly conserved patterns of evolution (small values of omega) rather than additional evidence for lineage-specific positive selection. In other words, whether selection for big brains translates to evidence for positive versus purifying selection on the sequence (i.e., the value of omega) would seem to depend on ancestral state. It would be helpful to readers if you can outline your hypotheses and predictions here more explicitly, including your expectations related to lineage-specific rates.

2. On a related note, I appreciate the attempt to find a matched set of control genes for the protein molecular evolution analysis; clearly this is difficult. However, I am not entirely convinced that the control genes are an appropriate background set (although I am also not sure how to identify one). Because one of the main filters for identifying these genes was to find genes where homologues can be readily identified, without the additional curation/sequencing done for TRNP1, they are likely to be enriched for genes evolving under constraint within mammals-so they may control for demographic differences, but not for the interaction of Ne with positive selection. This appears to be the case based on Figure 1—figure supplement 3, where most genes show no evidence of sites under positive selection. The comparison therefore works as a null for testing for positive selection, but not as an ideal null for testing whether patterns of positive selection correlate with a brain size/gyrification, as intended here (since most genes don't show the same pattern of positive selection).

3. In Figure 1 C and D and lines 104-105, the strongly attenuated slope in the control genes line is because you are taking the average across control genes for each species. The comparison I think you want is whether the slope for TRNP1 is an outlier relative to the slopes calculated for each of the control genes, not relative to the slope calculated for their average, which will inevitably be attenuated. I think this is the result you report in lines 106-109, but it should also be what is visualized in Figure 1C-1D to avoid overemphasizing the difference between TRNP1 and the control gene distribution (by taking the average, the distribution is not apparent). On a related note: are the results in these figures and text controlled for phylogenetic non-independence? I ask in part because it seems like a lot of the signal is being driven by haplorrhine primates.

4. You show, in a very nice result, that neural stem cell proliferation rate is correlated with brain size and gyrification index. Given that your evidence for the protein evolution-trait link is based on lineage-specific omega, is proliferation rate therefore correlated with omega?

5. Line 189 (and related to the question below about the directional relationship between TRNP1 expression and brain morphology) draws on the MPRA results to argue that the expression regulation of TRNP1 co-evolves with cortical folding. Is TRNP1 indeed more highly expressed in catarrhines with higher GI values (i.e., is this result consistent with a prediction from the literature?).

*Reviewer #3 (Recommendations for the authors):*

I would suggest that the authors address the following points:

1. Protein evolution rate analyses: I found Fig1S3 surprising, as it seems that about half of all control proteins used by the authors also evolve under positive selection. This seems unlikely high, and suggests that either the alignments contain errors and require cleaning, or the false discovery rate is inadequately corrected for in this analysis. Additionally, it was not always clear to me when the authors use either branch or site tests – I assume that the evolutionary rate analyses use unconstrained branch models, but this does not appear in the methods. In this case, can branch length confound the signal? I would expect that the estimation of omega in human, for example, is less reliable than e.g. dolphin, where the terminal branch is much longer.

2. Throughout the paper, it is unclear how multiple testing was corrected for. In some cases this does not matter, but it does for example when the authors investigate correlations between CRE activity and brain phenotypes at multiple sites. The likelihood of finding one spurious correlation increases rapidly when multiple CREs are tested, and in this case the authors cannot fall back on control regions to estimate the probability of observing such a correlation from background data, as they do for the gene evolution analyses. It may not be possible to control for this, but this should be explicitly acknowledged, and the conclusions toned down as a consequence.

3. I do not think it is surprising that an enhancer active in brain contains an excess of binding sites for TFs involved in neuronal proliferation, especially as I don't fully understand the display of Figure 4C (what does "Fisher's p" mean? P-value of Fisher's exact test – which would mean that these enrichments are not actually significant? What does "significant/annotated" mean?). The evidence that CTCF binding sites are stronger in catarrhines is weak – again, it is not clear to me how multiple testing was corrected for here, and the probability of spuriously finding one TF with a correlation out of 22 is high. I would suggest discussing this more explicitly, and toning down the discussion on CTCF as I am unconvinced that this signal is specific.

4. Spotted a few typos that need correcting e.g. line 111 "showed higher a significant"; line 122 "orthologoues"; also some in the methods which should be caught by a spellcheck editor.

---

## [Author Response]

Essential revisions:1) The reviewers appreciated the effort to investigate both coding and regulatory changes and feel the MPRA study was well-designed. However, because the results of the MPRA are somewhat unclear (and necessarily incomplete as a test of regulatory function), please temper your conclusions accordingly. Additionally, clarify how you dealt with the multiple testing burden.

We thank the reviewers for their constructive assessment. We agree with the mentioned caveats regarding the MPRA results and have improved the manuscript accordingly: we have tempered our conclusions in the abstract, the results and the discussion. We also have made clear how we deal with the multiple testing burden and have added additional analyses. We think that our interpretation of the MPRA data is now much more transparent. Specifically, the changes include the following points:

We rewrote the relevant result section and describe the statistical testing more clearly, explicitly acknowledge the multiple testing burden and explain why additional evidence makes our MPRA results interesting (“*Thus, while the statistical evidence from our MPRA data alone is limited, we consider the GI association in catarrhines together with the additional evidence from Reilly et al. strong enough to warrant a more detailed analysis of the intron CRE.*”)To corroborate our findings regarding the association of the intron CRE activity with GI across catarrhines, we now generated a null-distribution by permuting all other CRE activities (231) and plugging these in the same PGLS setting. Only 8/1000 (0.8%) of the permuted activities showed similar association strength (LRT p-value<=0.003).As suggested by Reviewer 1, we have added that we find TRNP1 to have a significantly lower expression in cynomolgus macaque than in human neural progenitor cells (new Figure 4—figure supplement 1), providing additional evidence for our interpretation of the MPRA data.To validate a possible role of CTCF for the regulation of TRNP1 activity, we gathered available experimental data from a ChIP-seq experiment in human neural progenitor cell lines, information about topologically associated domains (TADs) and histone marks for human gestational zone in the relevant developmental phase (gestational weeks 7-8 that contain the relevant neural progenitor cells). This data confirms CTCF binding in the intron CRE in vivo and further supports a role of this element in brain development (Figure 4—figure supplement 2).

2) Please address the outstanding questions about the appropriateness of the background set of genes used for comparison of brain phenotype-evolutionary rate correlations. This could take the form of additional analysis, or minimally a discussion of the potential limitations and/or biases of the set you used.

We think that a large part of the outstanding questions come from a misunderstanding of our analysis of TRNP1 protein evolution due to a suboptimal description on our side. We have completely restructured the relevant result section, revised Figure 1 and its supplements, so that in this new version the logic of each analysis step is made explicit. With this extensive re-write we also describe the appropriate scope of the use of our set of control proteins. We provide specific answers to the reviewers concerns below, but would like to make the following points here that we also tried to stress in the revised manuscript:

We now explain in more detail the creation of the control protein set in the main text. In brief, we took all genes we could find that – as TRNP1 – have only one coding exon, have a similar length and full length orthologous sequences in the 30 analyzed species. We manually inspected the alignments and find, that they match TRNP1 with respect to total tree length, alignment quality and average omega (Figure 1—figure supplement 3). Furthermore, because our control proteins are matched in size and total tree length, the statistical power to detect an association between omega and the rate of brain size/ GI evolution should be comparable. So neither from our selection procedure nor from their omega distribution, there is a reason to assume that the control genes are biased.There is good reason to assume that all investigated phenotypes are indirectly linked to the effective population size (Ne), and thus to the efficacy of selection. This would induce a positive correlation between the phenotypes and omega. The main purpose of the control proteins here is to quantify the effect that variation in the global efficacy of selection has on our Coevol correlation estimates. Although, as expected, there is a slight positive correlation between changes in brain size/gyrification for all control proteins, the one for TRNP1 is stronger and thus must have additional causes. That TRNP1 is among 4.0% and 6.4% of the most correlated proteins further strengthens this view and suggests that even though TRNP1 probably was relatively important for mammalian brain evolution, many other proteins also played a role.The PAML site model identifies eight codons in TRNP1 that evolve under positive selection and hence are worth further functional investigations, but does not indicate brain-related TRNP1 evolution by itself. We hope that placing this analysis now after the decisive Coevol analysis helps to separate these two different aspects of protein evolution analysis.

Reviewer #1 (Recommendations for the authors):I think this paper makes an important contribution to the literature. Genomic analyses that control for phylogenetic context/non-independence remain rare (at least in primates), and the integration of functional genomic analysis and experimentation is an important strength. I have a few concerns, however:1. I am unclear on your interpretation of the correlation between brain size/gyrification and the rate of protein evolution-specifically, why a positive correlation should necessarily be expected. While comparative analyses often analyze correlated evolution between two traits (e.g., brain size and diet), this analysis seems more like analyzing the correlation between brain size and lineage-specific change in diet. If we assume that TRNP1 is a major driver of brain size, and selection on brain size fully accounts for selection on the gene, then the big-brained descendants of a big-brained ancestor might show strongly conserved patterns of evolution (small values of omega) rather than additional evidence for lineage-specific positive selection. In other words, whether selection for big brains translates to evidence for positive versus purifying selection on the sequence (i.e., the value of omega) would seem to depend on ancestral state. It would be helpful to readers if you can outline your hypotheses and predictions here more explicitly, including your expectations related to lineage-specific rates.

We are sorry that this has not been clear in the previous version. We have now rewritten this section extensively to clarify our expectations and what the Coevol algorithm is testing.

Briefly, the Coevol algorithm takes the observed trait (e.g. brain size) from extant species, reconstructs the ancestral states and infers the change that has occurred on all branches. Thus, if a big brained descendant species is inferred to have had a big brained ancestor, the phenotype-based estimate of the branch length will be short. Then, Coevol correlates the phenotype-based branch length with the omega estimate for that branch. Hence, Coevol tests the correlation of phenotype change and protein sequence change and also takes uncertainties of the estimates into account. It is capable of detecting both positive as well as negative correlations. A positive correlation, as we find it here, suggests positive selection on an increase in brain size, while a negative correlation would suggest selection on a decrease.

2. On a related note, I appreciate the attempt to find a matched set of control genes for the protein molecular evolution analysis; clearly this is difficult. However, I am not entirely convinced that the control genes are an appropriate background set (although I am also not sure how to identify one). Because one of the main filters for identifying these genes was to find genes where homologues can be readily identified, without the additional curation/sequencing done for TRNP1, they are likely to be enriched for genes evolving under constraint within mammals-so they may control for demographic differences, but not for the interaction of Ne with positive selection. This appears to be the case based on Figure 1—figure supplement 3, where most genes show no evidence of sites under positive selection. The comparison therefore works as a null for testing for positive selection, but not as an ideal null for testing whether patterns of positive selection correlate with a brain size/gyrification, as intended here (since most genes don't show the same pattern of positive selection).

Initially, a bias towards higher conservation in our set of control proteins was also one of our biggest worries. However, the fact that the distribution of our omega estimates and total tree length is not biased and that TRNP1 falls somewhere in the middle of this distribution has alleviated this concern. All these checks as well as the selection process of the proteins is now described in detail in the main part of the results.

Moreover, we agree that the efficacy of selection boosts both purifying and directional (positive) selection. However, they have opposite effects on omega. Under purifying selection such a boost would decrease omega, while it would increase under positive selection as the dominant force. Because purifying selection is much more prevalent, it is unlikely that we detect the overall relatively mild effects due to Ne modulated variation in the efficacy of selection due to positive selection.

As a side note, we want to point out again that we extensively restructured the results part of the protein selection analysis. We mainly use the PAML site model to find which part of TRNP1 protein has been the target of selection to facilitate further functional analysis. Hence the results of the site model for the control proteins are not very relevant and were rather distracting from the real evidence. Therefore, we removed the old Figure 1 —figure supplement 3.

3. In Figure 1 C and D and lines 104-105, the strongly attenuated slope in the control genes line is because you are taking the average across control genes for each species. The comparison I think you want is whether the slope for TRNP1 is an outlier relative to the slopes calculated for each of the control genes, not relative to the slope calculated for their average, which will inevitably be attenuated. I think this is the result you report in lines 106-109, but it should also be what is visualized in Figure 1C-1D to avoid overemphasizing the difference between TRNP1 and the control gene distribution (by taking the average, the distribution is not apparent). On a related note: are the results in these figures and text controlled for phylogenetic non-independence? I ask in part because it seems like a lot of the signal is being driven by haplorrhine primates.

We agree with the reviewer that the distribution of the control genes is very relevant and we have now changed Figure 1 to also show this distribution of partial correlations across control proteins relative to TRNP1 (G, H). As these are calculated as for TRNP1 by Coevol, they are also controlled for phylogenetic non-independence. This analysis stresses the relative importance of TRNP1 for brain evolution, however, the average lines that we show in panels D-F show that the detected correlation of TRNP1 cannot be explained by Ne effects alone.

4. You show, in a very nice result, that neural stem cell proliferation rate is correlated with brain size and gyrification index. Given that your evidence for the protein evolution-trait link is based on lineage-specific omega, is proliferation rate therefore correlated with omega?

Yes, given the agreement in the direction of all considered associations, we see it as implied. However, as we assayed only 5 constructs we chose to investigate the main relationship of interest (proliferation vs. traits) directly.

5. Line 189 (and related to the question below about the directional relationship between TRNP1 expression and brain morphology) draws on the MPRA results to argue that the expression regulation of TRNP1 co-evolves with cortical folding. Is TRNP1 indeed more highly expressed in catarrhines with higher GI values (i.e., is this result consistent with a prediction from the literature?).

We thank the reviewer for this suggestion. Indeed, as expected we find that TRNP1 has a significantly lower expression in cynomolgus macaque than in human neural progenitor cells. We added this in Figure 4—figure supplement 1A-C and mention it in the main text (line 264): “Moreover, TRNP1 expression was significantly higher in human NPCs than those of cynomolgus macaque’s (BH-adjusted p-value<0.05, Figure 4—figure supplement 1A-C), consistent with higher intron CRE activity.”

Reviewer #3 (Recommendations for the authors):I would suggest that the authors address the following points:1. Protein evolution rate analyses: I found Fig1S3 surprising, as it seems that about half of all control proteins used by the authors also evolve under positive selection. This seems unlikely high, and suggests that either the alignments contain errors and require cleaning, or the false discovery rate is inadequately corrected for in this analysis. Additionally, it was not always clear to me when the authors use either branch or site tests – I assume that the evolutionary rate analyses use unconstrained branch models, but this does not appear in the methods. In this case, can branch length confound the signal? I would expect that the estimation of omega in human, for example, is less reliable than e.g. dolphin, where the terminal branch is much longer.

We rewrote the results part “TRNP1 amino acid substitution rates co-evolve with rates of change in brain size and cortical folding in mammals” to clarify the logic that led to our main conclusion. To begin with, we now also describe our quality checks of the alignments in detail. Briefly, all multiple sequence alignments were manually checked, none of them contain excessive amounts of missing data and we removed one protein alignment because it was an outlier with respect to the total tree length. We also provide all multiple alignments on the accompanying github-page https://github.com/Hellmann-Lab/Co-evolution-TRNP1-and-GI (Figure 1 —figure supplement 3B). In the old Fig1S3, we plotted the results of the PAML site model comparison between 2 modes of evolution (M8 vs M7), which is not a very stringent model and was originally only intended to prioritize sites within TRNP1 for future functional analysis. We now removed this misleading plot and only discuss the site model within the scope of our original intent.

The analysis that lead us to conclude that TRNP1 protein coding evolution is linked to the evolution of brain size, is based on Coevol, “A Phylogenetic Model for Investigating Correlated Evolution of Substitution Rates and Continuous Phenotypic Characters” (Lartillot and Poujol, MBE, 2010). Briefly, Coevol is a Bayesian MCMC that optimizes a co-variance matrix that includes the branch lengths (λ_S_), ω and the continuous traits, using a Brownian motion process. Hence, the variation of the reliability due to branch lengths is explicitly factored into the model and does not bias our estimates of co-evolution.

2. Throughout the paper, it is unclear how multiple testing was corrected for. In some cases this does not matter, but it does for example when the authors investigate correlations between CRE activity and brain phenotypes at multiple sites. The likelihood of finding one spurious correlation increases rapidly when multiple CREs are tested, and in this case the authors cannot fall back on control regions to estimate the probability of observing such a correlation from background data, as they do for the gene evolution analyses. It may not be possible to control for this, but this should be explicitly acknowledged, and the conclusions toned down as a consequence.

We agree on this point as also laid out in the essential revisions above. The observed signal across all mammals is indeed weak, especially given the multiplicity of testing. This might be in part due to our inability to reliably detect and assay the full (functionally) orthologous regulatory sequences in species that are phylogenetically far from the human. This can be seen in the reduced orthologous CRE length beyond primates (Figure 3 —figure supplement 1). And indeed, as you state, generating data for a large number of control regions is difficult. To generate a type of a null-distribution for the observed correlation across catarrhines, we have permuted the activities of all other CREs of this data set and only 8/1000 (0.8%) show higher or equal significance in the association. So we think we have now explicitly acknowledge these limitations, the multiple testing issue, gathered additional corroborating evidence e.g. from Reilly et al. and toned down our conclusions e.g. by saying:

“Moreover, we find tentative evidence that the activity of a regulatory element in the intron of TRNP1 might be associated with gyrification in catarrhines.” (line 294)

“In summary, we find a suggestive correlation between the activity of the intron CRE and gyrification in catarrhines, indicating that also regulatory changes of TRNP1 might have contributed to the evolution of gyrification.” (line 286)

3. I do not think it is surprising that an enhancer active in brain contains an excess of binding sites for TFs involved in neuronal proliferation, especially as I don't fully understand the display of Figure 4C (what does "Fisher's p" mean? P-value of Fisher's exact test – which would mean that these enrichments are not actually significant? What does "significant/annotated" mean?). The evidence that CTCF binding sites are stronger in catarrhines is weak – again, it is not clear to me how multiple testing was corrected for here, and the probability of spuriously finding one TF with a correlation out of 22 is high. I would suggest discussing this more explicitly, and toning down the discussion on CTCF as I am unconvinced that this signal is specific.

We have described this analysis better in the text and changed the axis label in Figure 4C to “intron-binding expr. TFs/expr. TFs” (see also response to a minor point of reviewer 1).

This plot summarizes a gene-set enrichment analysis on GO categories, where we only consider expressed TFs in NPCs and look for categories with an enrichment of TFs that also have a binding site in the intron CRE. To test the significance of such an enrichment, we use topGO (Rahnenfuehrer, Bioinformatics, 2006) with a prioritization of more specific categories (elimination algorithm) and the reported p-values are indeed from a Fisher’s-exact test. Because ontology groups are inherently inter-dependent, multiple testing corrections would be overly stringent and hence are not commonly applied in this context.

We are unsure how surprising it is to find an enrichment of TFs involved in particular Gene Ontology categories like neuronal proliferation, but we think it is relevant enough to be reported, also because TRNP1 is expressed in other tissues, such as the stomach, liver or kidney; and, as mentioned, we only use NPC-expressed TFs as the background.

Regarding the binding sites, we agree that testing 22 TF binding site variation across only 10 species is underpowered. We stress now that this analysis serves to pinpoint the best candidates and explicitly states that the p-value is uncorrected. To strengthen the possible involvement of the strongest candidate CTCF, we have gathered experimental ChIP-seq and Topologically Associated Domain (TAD) data from relevant cell types (Figure 4 —figure supplement 2, lines 277-282), showing that intron CRE is indeed bound by CTCF, which is associated with a nearby TAD boundary. Hence, this CTCF binding site indeed seems functional and the binding strength could affect the binding of other TFs in vivo and in vitro, and TRNP1 expression and TAD boundaries in vivo. But we agree that this is still not conclusive and requires experimental evidence to really understand its potential role (as stated in the discussion).

4. Spotted a few typos that need correcting e.g. line 111 "showed higher a significant"; line 122 "orthologoues"; also some in the methods which should be caught by a spellcheck editor.

Thanks, we corrected this.